# Hard, Soft, and Hard-*and*-Soft Drug Delivery Carriers Based on CaCO_3_ and Alginate Biomaterials: Synthesis, Properties, Pharmaceutical Applications

**DOI:** 10.3390/pharmaceutics14050909

**Published:** 2022-04-21

**Authors:** Yanqi Huang, Lin Cao, Bogdan V. Parakhonskiy, Andre G. Skirtach

**Affiliations:** NanoBio Technology Group, Faculty of Bioscience Engineering, Ghent University, 9000 Ghent, Belgium; yanqi.huang@ugent.be

**Keywords:** calcium carbonate, alginate hydrogels, drug delivery, drug loading, release mechanism

## Abstract

Because free therapeutic drug molecules often have adverse effects on normal tissues, deliver scanty drug concentrations and exhibit a potentially low efficacy at pathological sites, various drug carriers have been developed for preclinical and clinical trials. Their physicochemical and toxicological properties are the subject of extensive research. Inorganic calcium carbonate particles are promising candidates as drug delivery carriers owning to their hardness, porous internal structure, high surface area, distinctive pH-sensitivity, low degradability, etc, while soft organic alginate hydrogels are also widely used because of their special advantages such as a high hydration, bio-adhesiveness, and non-antigenicity. Here, we review these two distinct substances as well as hybrid structures encompassing both types of carriers. Methods of their synthesis, fundamental properties and mechanisms of formation, and their respective applications are described. Furthermore, we summarize and compare similarities versus differences taking into account unique advantages and disadvantages of these drug delivery carriers. Moreover, rational combination of both carrier types due to their performance complementarity (yin-&yang properties: in general, yin is referred to for definiteness as hard, and yang is broadly taken as soft) is proposed to be used in the so-called hybrid carriers endowing them with even more advanced properties envisioned to be attractive for designing new drug delivery systems.

## 1. Introduction

The internal microenvironment of the human body is a complex circulatory system with some exclusivity to foreign objects. This function protects the human body from viruses, while having the opposite effect on the treatment of diseases to some extent. One of the issues is the decomposition capability of enzymes in body fluids which would consume drugs before reaching the pathological site, thereby reducing the effective drug concentration and therapeutic effect. Simultaneously, the absorption of free drugs by normal cells or tissues during delivery will impose serious adverse effects to human body, which is a widespread problem with current chemotherapeutic drugs and in the area of nanomedicine in general [1].

In view of this, various multifunctional drug delivery carriers capable of encapsulating therapeutic drugs molecules have emerged or have been developed. These carriers allow the drug to selectively kill abnormal cells without adverse effects on normal cells, thus minimizing side effects on the human body. Numerous drug delivery carriers with favorable biological properties have been developed, including carriers belonging to two main classes: inorganic and organic. The former one, inorganic particles, includes carriers such as quantum dots (QD), gold (Au) and silver (Ag) metal nanoparticles, clays, carbon, porous silica (SiO_2_), hydroxyapatite (HA), and calcium carbonate (CaCO_3_), etc. The latter one, organic particles, comprises of an equally impressive range of carriers, such as liposomes, polymer micelles, dendrimers, protein conjugates, red blood cells (RBCs), cell penetrating peptides, layer-by-layer (LbL) capsules and alginate (Alg) or alginate hydrogel (Alg-Hs) particles, etc. (Figure 1). In this review, we describe one candidate from each category: CaCO_3_, an inorganic carrier, generally referred to as hard, and Alg, an organic carrier, generally referred to as soft; a combination of CaCO_3_ on Alg and Alg in CaCO_3_, generally referred to as hard-*and*-soft or hybrid [2], is also discussed. Peculiarly, although they are quite different, there are similarities and, naturally, dissimilarities between these types of carriers. However, first we give a short overview of major types of both inorganic and organic drug delivery carriers.

Numerous inorganic particles have been widely explored in the field of drug delivery carriers because of their respective advantages. The small size and versatile surface chemistry of QD allow incorporating them within virtually any nano drug carriers with minimal impact on the overall properties and provide excellent optical properties, which are useful in areas such as drug delivery, sensors, and bioimaging. However, several issues still need to be addressed in order to realize clinical applications of QD, such as overall toxicity, body clearance, scalability of synthetic protocols, environmental impact, and manufacturing costs [3,4]. Au and Ag nanoparticles have high specific surface areas, functionalized surfaces, adjustable optical properties, and strong sensitivity to Raman signal, which allow them to be widely used in drug carriers, imaging, radiotherapy enhancement, and other fields [5,6,7]. However, these noble metal nanoparticles have potential problems such as high cost, somewhat extensive equipment requirements for physical preparation methods, un-environmental chemical preparation, and a potentially strong nano-agglomeration effect, so they are mostly used as doping phases to improve the performance of carriers. Excellent physicochemical properties and biocompatibility of clays, especially a high porosity and adsorption capability of nano-clays, as well as the controlled release of drug carrier systems provided by their interoperability with drugs, strongly support their applications in drug delivery system. while systematic studies need to be performed on the effectiveness, specificity, pharmacokinetics, toxicity, and safety of these drug delivery carriers [8,9]. Furthermore, the recalcitrance and long-term toxicity of carbon are the key issues limiting their applications in human body [10]. SiO_2_ particles, particularly in their mesoporous form, possess intrinsic hydrophilicity, biocompatibility, and their excellent mechanical properties prevent the core-shell structure and internal cargo from being damaged [11,12]. Micro-nano SiO_2_ particles prepared by the sol-gel method have a preferable blood stability compared with other particles, such as lipids and polymeric particles. However, the existence of targeted receptors will affect the original hydrophilicity of SiO_2_, necessitating further research in the field of targeted drug delivery [11]. Kester et al. [13] adopted a double reverse micelle strategy to synthesize amine, carboxylate, and poly(ethylene glycol) (PEG) surface functionalized calcium phosphate (Ca_3_(PO_4_)_2_) nanocomposite particles, and encapsulated a fluorophore and hydrophobic anti-tumor drug ceramide. This carrier system remained stable in 37 °C physiological solutions, and drug molecules were released by the dissolution due to pH changes during endocytosis, but the chronic toxicology of such Ca_3_(PO_4_)_2_ nanoparticles still needs to be verified, since an increased toxicity was noticed with an increase of their concentration [13]. As one of the bone components, HA has been applied in biomaterial science and tissue engineering due to its biocompatibility and potential medical therapeutic avenues, whereas it is still necessary to understand how HA nanoparticles interact with a biological environment (cell, tissue, organ, or system). And it is important that they will not be just thought of as a structure that will be accumulated, internalized, or used to deliver a cargo, but a construct that will directly or indirectly modulate cell signaling pathways to improve or worsen a specific function, through the cell/HA functional groups contained in both [14].

As a commonly used inorganic substance, calcium carbonate particles (CCPs) possess several advantages compared with other biological and biomimetic materials. On the one hand, the raw materials for synthesizing CCPs are easily available and less expensive, while the preparation process of CCPs is facile, safe, and eco-friendly, which does not hinder the achievement of large-scale reproducible production. On the other hand, CCPs are relatively hard and have a high porosity, large surface areas, favorable biocompatibility, pH-sensitive response, slow degradability, and low-cytotoxicity, etc. The advantages mentioned above are sufficient to make CCPs great promising materials for applications in the field of biomedicine, especially for drug delivery and targeted therapy (Figure 1). One example, where mechanical properties of carriers or hardness are important is intracellular delivery, where carriers have to withstand at least 0.2 μN—pressure exerted by cells upon uptake [15].

Organic vesicles and particles represent perhaps even a larger class of drug delivery carriers, where liposomes are some of the first and still very broadly used types of carriers. They are formed by phospholipid bilayers and possess a good biocompatibility, a good targeting capability, and a structure similar to that of biological cells, whereas the carriers have a low encapsulation rate for some water-soluble substances, poor on-shelf stability and repeatability, hurdles in scale-up and costs in commercialization processes [16]. Polymeric micelles exhibit a high solubility, excellent loading capacity, high stability in blood, and long lifespan, which can be linked by an unique core-shell structure, and the hydrophobic part formed through the physical or chemical binding mode is capable of encapsulating hydrophobic drugs and proteins. The brush-like structure of the hydrophilic part allows it to protect the hydrophobic and ensures its delivery [17]. However, the efficient delivery of drugs from polymer micelles, the kinetics of drug release, and the degradation of carriers substances in blood and their clearance need to be further studied [18,19]. Dendrimers can render drugs greater water solubility, bioavailability and biocompatibility, and are compared to traditional macromolecules. But dendrimers have a higher density of surface functional groups, which allows a higher drug loading efficiency, but toxicologic status of candidate dendrimers requires further investigation [20,21]. Protein conjugates combine materials science, chemistry, and biology allowing the design of structural designability and advanced polymer properties, while new techniques need to be developed to integrate biological units, and the structures of many polymers [22]. Another class of perfectly biocompatible drug delivery carriers is RBCs. Even though RBCs exhibit an excellent accessibility and biocompatibility, tunable drug loading capacity and long blood circulation half-life, some critical issues should not be ignored in application of RBC-based drug delivery systems [23]. For instance, RBCs are vulnerable to osmotic stress and excessive cross-linking, which reduces their elasticity and deformability, which, in turn, causes a high risk of complement activation, triggers poor clearance of RBCs by reticuloendothelial system, and shortens their circulation period in blood. Secondly, cryogenic and anaerobic storage conditions are essential for preserving the quality of RBCs. Otherwise, they are susceptible to storage lesion, leading to enhanced clearance and immune response regulation. Additionally, RBC cannot penetrate and cross tissue barriers except within the vascular space, and the leakage of loaded molecules may cause side effects [23]. Cell penetrating peptides can facilitate the uptake of large, biologically active molecules into mammalian cells, but the immunogenicity and long-term side effects still require much investigation [24]. It is also worth mentioning another drug delivery system—polyelectrolyte capsules, which are produced by the LbL adsorption method. For such capsules, there are various methods are developed for encapsulation and release, including the remotely activated release methods [25,26,27]. Capsules can circulate, remain in the circulation system, but larger (micrometer) microcapsules eventually accumulate in the liver and kidneys [27]. However, the interaction of such capsules with living systems requires further investigation.

Alg is a natural unbranched polysaccharide with a non-toxic, non-antigenicity, good biocompatibility and biodegradability, high flexibility and elasticity, low density, and pH stability, having many different applications both in the field of pharmaceutical and biomedicine [28,29]. As polyanionic compounds, Alg possesses one of the most important properties that is used to readily form hydrogels with high water absorption, while keeping their network structure through ionic crosslinking, chemical crosslinking, free radical polymerization and phase transition [30,31] (Figure 1). Those hydrogels can sense subtle changes from external stimuli such as temperature, pH, ionic concentration, electricity and magnetism, and modulate their swelling state. Therefore, Alg-hydrogels (Alg-Hs) delivery system has been widely used for protecting and loading bioactive substances, drugs, genes and cells [32].

In this review, we describe CCPs and Alg-Hs as representatives of two opposite classes of materials (inorganic and organic, i.e., hard and soft, respectively) highlighting their similarities and differences as well as advantages and disadvantages. We look at them as both stand-alone carriers and a combination resulting in a hybrid carrier combining these materials. The analysis here is made looking at the intrinsic properties to the preparation methods (green and facile) and performance in biological and pharmaceutical applications. In addition, the cross-linking of Alg can be induced by Ca^2+^, and the diverse combination of CCPs and Alg can form hybrid carriers with coatings, hollow Alg capsules as well as CCPs-enhanced Alg-Hs—all of which make these two materials even more attractive. Therefore, this review summarizes fundamental properties and mechanisms of the formation, synthetic methods, and applications of CCPs and Alg-Hs as carriers for drug delivery and targeted therapy. Moreover, the drug loading efficiency, release mechanisms and applications through different administration modes are emphasized and compared, thus providing insights into improving performance of these drug delivery systems with a perspective on their broader exploitation in biomedicine.

## 2. CaCO_3_ Carriers

### 2.1. Characteristics and Properties of CaCO_3_ Particles

CaCO_3_ is a naturally occurring inorganic mineral that exists of amorphous CaCO_3_ (ACC) and three polymorphs: calcite, aragonite and vaterite (Figure 2) [33,34]. Generally, ACC exists as hydrated CaCO_3_, like CaCO_3_·6H_2_O and CaCO_3_·H_2_O, with a high solubility and plasticity, but no established crystal structure exists [35,36]. Among these three polymorphs, calcite is the most stable phase with a rhombohedral structure that belongs to the hexagonal crystal system, and it tends to form blocks in natural world owing to the agglomeration of particles, which is determined by its kinetics [36,37,38]. Aragonite is a metastable phase with a needle-shape morphology, and it mainly exists in the shells of mollusks [36,37]. The crystal structure of vaterite is the hexagonal crystal system with spherical or elliptical morphologies. Vaterite possesses the worst stability among the three phases and is easily transformed into calcite in an aqueous solution [39,40]. According to the phase transition kinetics and mechanism of CaCO_3_, the crystallization from ACC to CaCO_3_ can be divided into two stages: (a) the dehydration of ACC to form vaterite; and (b) the vaterite is converted to calcite through dissolution and reprecipitation mechanisms, where the reaction rate is controlled by the surface area of calcite [36,41]. In addition, experimental parameters including temperature, pressure, pH of the solution, stirring method, reaction time, ion concentration and ratio, and the additives are the key factors affecting phase transition, which makes the synthesis of CCPs with desired morphologies much more achievable [42,43,44,45]. Noticeably, although vaterite is unstable, its spherical structure, high specific surface area, porosity, solubility, and phase-transition properties facilitate the adsorption and release of cargoes, making it a potentially attractive drug delivery carrier.

### 2.2. Synthesis of CaCO_3_ Particles

A large variety of methods have been adopted for the synthesis of CCPs, including CaCO_3_ microparticles (CCMPs) and nanoparticles (CCNPs), which can be classified into two principal categories: (a) the solution route [45,46,47,48,49]; and (b) the carbonization route [50,51,52]. The core principle of the former synthesis method is the reaction of Ca^2+^ and CO_3_^2−^ ions in an aqueous solution to form insoluble precipitate, as shown in Figure 3 [53]. It has been identified that water [54], ions [55], and biomolecules [56] play an important role in the formation process of calcium carbonate. In regard with water, the mechanism of different routes shown in the Figure 3 is as follows: (i, ii) when solutions are undersaturated with respect to ACC but supersaturated with respect to any other crystalline CaCO_3_ phases, the crystalline nuclei will bypass ACC and grow into larger crystals through diffusion and adsorption of Ca^2+^ and CO_3_^2−^ ions [57]. Of note, in most cases, ACC can be preferentially formed through different mechanisms [58]; (iii) counter ions combined with each other to form highly hydrated chains of ion clusters, the so-called prenucleation clusters (PNCs), subsequently aggregate to form a liquid phase or small ACC particles [59]; (iv) aqueous solutions containing Ca^2+^ and CO_3_^2−^ ions undergo a spinodal or binodal liquid–liquid phase separation to form a solute-rich liquid phase, in which ACC particles are formed through dehydration or nucleation of the liquid phase [60]; (v, vi, viii) ACC nuclei are directly formed in an aqueous solution, evolving then into smaller ACC particles and growing until crystals are formed, or (vii) dissolve and recrystallize in contact with water.

The former route is conducted in Ca^2+^-H_2_O-CO_3_^2−^, or Ca^2+^-R-CO_3_^2−^ reaction system (R is an organic medium), where the solutions containing Ca^2+^ and CO_3_^2−^ are mixed under specified conditions, and then white CCNPs are precipitated owing to the solubility differences [42,43,49]. While in the latter route, an aqueous emulsion of Ca(OH)_2_ is utilized as a calcium source and carbonized with CO_2_ to synthesize CCNPs: this approach includes microemulsion method, biomineralization method, etc. [50,61,62]. Although different researchers have reported preparation of CCPs (to some extent those approaches are different in the choice of raw materials, equipment for synthesis, or even mineralization in some organisms), the core principles are inseparable from the above two routes. Dadkhah et al. [48] used calcium nitrate as raw material via a thermal treatment and solid-state method to synthesize CCNPs, and different characteristic methods indicated that the obtained particles were calcite with a cubic morphology. In addition, CO_3_^2−^ can be obtained by dissolving CO_2_ or CO_2_-carrying substances in an aqueous solvent. After that, stable CCNPs can be obtained by injecting these specific gases into a Ca(OH)_2_ solution at a certain flow rate, because a large amount of Ca^2+^ ions in Ca(OH)_2_ solution can inhibit the re-dissolution of CaCO_3_ during the reaction process [51,62]. Wang et al. [50] utilized the carbonization of Ca(OH)_2_ slurry to synthesize hydrophobic CCPs with an active ratio of up to 100%, and the existing oleic acid in the slurry played a role in controlling the particle growth and modifying the surface of the particles. Yang et al. [61] synthesized CCPs with two phases (vaterite/calcite) using one-pot, l-lysine (Lys)-mediated biomineralization method, and Lys not only can control the morphology and phase distribution of particles, but also promotes the yield.

Moreover, considering available reagents, reasonable costs, and environmentally friendly raw materials, utilizing CaCl_2_ and Na_2_CO_3_ solutions to synthesize CCPs via chemical precipitation is attractive, for example, to control such parameters as size. In this regard, Parakhonskiy et al. [49] synthesized spherical CCPs with dimension of 0.4–4.5 μm in special solvents (water and ethylene glycol (EG)) by mixing the above two solutions using magnetic stirring with different reaction times at room temperature. The size of the particles was smaller with at a higher percentage of EG in the solvent, because EG slows down the reaction speed. Different agitating methods of mixed solution, such as ultrasonic and microfluidic, were investigated as well [43,46]. Inspired by the described above research, Sovova et al. [42] used the superiority of a special reactor, which can accurately control not only temperature, but also detect changes of ionic concentration of the solution during the reaction process to explore the influence of temperature (10–50 °C) and stirring time (5–120 min) on the morphology, size and structure of CCPs in aqueous solvents. It was also possible to obtain higher yields of the synthesis. It was found that vaterite gradually transformed to calcite with increasing both temperature and time, while the size of particles remained stable after 15 min meaning that the reaction was finished at that point. In addition, using temperature as a control mechanism, the essential property (porosity) of CCPs has been tuned in the nanometer range, and such nano-porosity control is particularly important for enzyme loading in CCPs [63,64]. Svenskaya et al. [44] adopted a dropwise precipitation to study the influence of the ratio of EG/water, reagent concentration, and the dropping rate on the size, shape, and relative content of vaterite of the synthesized CCPs. Results demonstrated that vaterite had a size range of 400 nm to 2.7 μm, especially the smallest particles were obtained in EG:H_2_O = 4:1 at the dropping rate of 0.084 mL/min at the lowest concentration of 0.05 mol/L. Interestingly, anisotropic and elongated CCMPs have been synthesized by admixing respective salt solutions with different concentrations of reacting ions (Ca^2+^ and CO_3_^2−^) [65], while the particle size was observed to be influenced by the relative ratio of these salts [66]. An interesting application of such anisotropic carriers has been to stimulate selective cell uptake [45]. In general, more advanced methods have been demonstrated and adopted to improve the properties of CCPs or CCPs-based hybrid (functionalized with Alg or polymers) system for different application, especially in the field of drug encapsulation [52,67,68].

### 2.3. Drug Loading and Release

#### 2.3.1. Drug Loading

In addition to such properties as biocompatibility, degradability, and low-toxicity, possessing a controllable porosity, hollow structure, and high surface areas is essential for efficient drug delivery carriers, since they can provide sufficient internal spaces and powerful attraction forces for the loading and storage of drug molecules [69]. To achieve the loading of drug molecules into CCPs, two methods can be adapted: physical adsorption and co-precipitation. It should be noted that the loading efficiency of carriers varies with the size and morphology, properties of drug molecules, loading time and method [12]. Some results from relevant studies are presented in Table 1.

##### Physical Adsorption Loading Method

Physical adsorption is a facile and feasible method implemented in an alternative solution according to the aqueous solubility of drug molecules. These molecules can be adsorbed into the pores and on the surface of CCPs through electrostatic forces and osmosis until equilibrium is reached via mixing the expected CCPs with the solution containing a predetermined concentration of drugs. Various factors influence on the loading efficiency of CCPs carriers, such as porosity and surface area of carriers themselves, molecule weight (MW) and surface charge of drugs, doping of polymers, the feeding concentration of drugs, and mixing time as well as the method of adsorption (Figure 4).

The size and shape of CCPs have a significant influence on their porosity and surface area, which in turn affects the loading of drugs [43,92]. For example, vaterite containers with two dimensions (650 ± 30 nm and 3.6 ± 0.5 μm) exhibited loading efficiencies of 1.4 ± 0.4% (*w*/*w*) and 0.9 ± 0.2% (*w*/*w*) to photosensitizers, respectively [73]. All adsorption properties are based on the dynamic equilibrium of the adsorption-desorption process. In this case, vigorous shaking or ultrasonic stirring can improve the probability of collision between particles, thus increasing the loading efficiency of carriers [43,93].

Normally, molecules with a relatively high MW can be loaded in greater amounts, because they leak less through permeation [43,94]. For example, Svenskaya et al. [43] investigated the loading efficiency of vaterite with different surface areas (25 m^2^/g and 10 m^2^/g) to substances with a relatively high MW (tetramethyl rhodamine isothiocyanate–bovine serum albumin (TRITC–BSA)) and a relatively low MW (model rhodamine 6G (Rh6G)). There, particles with a higher porosity and surface areas had higher loading capacity for high MW substance (9.7 ± 0.1 wt% compared to 7.5 ± 0.1 wt% for particles with a lower porosity and surface areas), while no significant difference was observed for low MW (0.15 ± 0.05 wt%) for any size of vaterite particles. This can be explained as follows: larger MW molecules were mostly adsorbed on the surface but smaller MV molecules had stronger penetration effect and thus depended to a lesser degree on the surface of carriers. Additionally, molecules with stronger opposite charges will be more easily absorbed [79]. The introduction of some polymers would change the electrostatic charge of CCPs, thus providing stronger adsorption capacity for loading of drug molecules [95,96]

The increase of the feeding concentration of drugs can positively affect the loading capacity of carriers, whilst their encapsulation efficiency will decrease [96,97,98]. The loading time is not simply positively correlated with the loading capacity of carriers. A too short time is insufficient for drug molecules to be fully adsorbed, but once the adsorption-desorption between the matrix and drug molecules reaches a dynamic equilibrium, the continuous increase of the time will not result in a larger loading capacity [75,99].

In the freezing process of the liquid phase, the thrust generated by the freezing at the solid-liquid interface combines with the various moving speeds of drug nanoparticles, and the crystallization front will squeeze former into the surface and pores of CCPs particles which have a much slower moving speed [100]. The repeated freezing and thawing process will achieve 3–5 times higher loading efficiency than that of the traditional method, while the effect of this method is highly related to liquid composition, particle composition and size, and freezing conditions.

##### Co-Precipitation Loading Method

The co-precipitation method is a simple and efficient way to synthesize desired drug carriers via a one-step reaction, which is conducted by dispersing drug molecules uniformly in a Ca^2+^ ions solution, after which these molecules are encapsulated into the interior or pores of CCPs when Ca^2+^ ions react with CO_3_^2−^ ions to form precipitates. During this process, drug molecules capable of reacting with one of the ions (Ca^2+^ or CO_3_^2−^) increase the loading efficiency but also result in an unpredictable size and porosity of the carriers. For example, Zhao et al. [101] utilized CaCl_2_ ethanol solution containing DOX to react with CO_3_^2−^ ions provided by ammonium bicarbonate to synthesize CCPs/DOX. In this case, DOX was chelated with Ca^2+^ ions, and the size of spherical CCPs/DOX was 130 nm with loading efficiency for DOX of 5.9%. Co-precipitation of DOX with DNA plasmids yielded 100 nm particles loaded with 1.29 wt% of DNA and 0.93 wt% of DOX [102].

The synthetic conditions of CCPs, such as the concentration of initial solution and drug molecules, types of solvent, doping of polymers, and reaction conditions play a decisive role in controlling the size, morphology and porosity of the resulting CCPs, which have been mentioned in Section 2.2, and these will ultimately affect the loading properties of CCPS [86,103,104]. So, these factors should be considered in the process of loading drug molecules via the co-precipitation method. Some examples of using co-precipitation include the following: Begum et al. [105] prepared tetracycline-loaded CCPs by co-precipitation, where the loading efficiency increased from 14.94 to 62.27 wt% with the increment of feeding concentration of tetracycline during synthesis. This indicated that an increment in the concentration of drugs in the mixed solution will increase the loading efficiency. A similar approach was adopted by Peng et al. [106] to load etoposide to prepare etoposide/CCNPs (2 μm) carriers, and the loading efficiency even reached 39.7%, because CCNPs had high surface area (82.14 m^2^/g) and an average pore size of 13.98 nm. Sudareva et al. [104] mixed various polyanionic polymers in different proportions into CaCl_2_ solution containing DOX, and synthesized the hybrid particles via coprecipitation method. This hybrid carriers showed a high drug loading efficiency, which can be explained by three aspects: (i) introduction of polyanions enhanced the adsorption of DOX cations by the carriers; (ii) polyanions changed the surface properties of CCPs; (iii) polyanions stabilized the CCPs and hindered their transformation from vaterite to calcite. Since the loading of drug molecules is carried out simultaneously with the synthesis of CCPs, the factors that can affect the formation of CCPs will affect their drug loading.

It should be noted that drug molecules loaded through physical adsorption are maintained by electrostatic force, and the adsorption efficiency is highly dependent on the characteristics of surface of both drug molecules and CCPs. Those molecules adsorbed by weak electrostatic forces will inevitably be desorbed and lost during the washing process, which will decrease the loading efficiency of carriers. However, this method can effectively maintain the original properties of drug molecules or the activity of proteins, while less resistance is present for the release of molecules [107]. In the co-precipitation method, the loading of molecules is driven by similar electrostatic interactions, but a higher loading efficiency can be obtained since molecules will be trapped in the voids during the formation process of CCPs, and there would be a lower leakage from carriers, even though a small number of drug molecules with small size will be missed [107]. However, the alkaline environment during particle formation process may reduce the activity of drugs, especially proteins, thus reducing drug efficacy [108]. Therefore, the polyelectrolyte shell formed by LbL assembly technique combined with the above loading method will provide an effective means for drug encapsulation into polyelectrolyte multilayer capsules.

#### 2.3.2. Release Mechanism

The carriers with a core-shell structure and sensitive responsiveness to external stimuli including pH, sound, light, heat, magnetism are desirable since they can induce targeted delivery and release of drugs [68,109]. CCPs, especially ACC and vaterite, as degradable carriers with a favorable porous structure, allow not only efficient loading of substances, but also permit achieving controllable release of drug molecules through the number of possible release mechanisms: (1) drug diffusion from the porous structure (Figure 5a–c), (2) transformation from ACC and vaterite, with a porous structure, to calcite with a non-porous structure (Figure 5a,b), (3) degradation of the CCPs matrix (Figure 5d,e).

In all cases, the diffusion mechanism is present because the encapsulated drugs have no chemical linkage with the calcium carbonate matrix but rather are related to weak Coulomb and Van der Waals forces. Noticeably, incubation in an aqueous solution with normal pH will lead to the recrystallization process of the unstable vaterite or ACC particles to the thermodynamically stable calcite phase, and the concomitant decrease in porosity will facilitate drug release [63,77,85]. All these processes strongly depend on the pH of the medium, and the particles dissolve quicker in a solution with acidic pH, but the shell of the calcium hydroxide can format in alkalic pH, which inhibits the dissolution of CCPs [73,74]. The presence of other ions can affect the recrystallisation rate, resulting in the release of drugs [49,75,110]. For example, positive ions like Na^+^ can neutralize the particle charges and stimulate the formation of particles aggregation and recrystallisation rate [49,75]. On the other hand, the presence of PO_4_^3−^ ions can stimulate the ionic exchange reaction to form a stable hydroxyapatite shell, which will prevent the following release of encapsulated molecules [49,75].

In another research, it was reported that vaterite containers with two dimensions (650 ± 30 nm and 3.6 ± 0.5 μm) loaded with photosensitizers demonstrated two release mechanisms: desorption and pH-induced phase transition [73]. Here, in the buffer solution at pH = 7, it took 1 day for the saturation of the release for larger particles, while it took 4 days for small particles. This is because the large particles completed the transition to calcite faster, which in turn promoted the release of the payload. In addition, the release saturation of large particles was reduced due to the strong resorption of calcite-forming particles, while small particles have a longer release period.

The loading of the photosensitizers demonstrated that decreasing the pH from 7.4 to 6.8 could significantly accelerate the release content from 30% to 50% in 3 h (Figure 5e) [74]. Similar dynamic demonstrated by DOX, for example, a low release rate (40%) of DOX was detected after 15 h of incubation in the phosphate buffer saline (PBS) at pH = 7.4, followed by a release rate of approximately 80% within 20 h. Decreasing pH to 4.8 stimulated the initial burst-release of up to 40% of encapsulated DOX within 15 min following with release of 60% of DOX after incubating for 30 min. A complete release was achieved after approximately 3 h. This different release mechanism provides a promising application in targeted release in tumor sites with an acidic microenvironment. Similar processes occurred to small CCNPs (42 nm) loaded with the hydrophobic anticancer drug docetaxel [109], which demonstrated that the rapid release of 84% and 45% occurred at pH = 4.8 and 7.4, respectively, in the first 3 h. After 72 h, it reached 90% at pH = 4.8 and only 60% at pH = 7.4. These results suggest that the drug-carrying system can effectively seal drugs during the plasma transport before reaching cancer sites [109].

#### 2.3.3. CaCO_3_ Functionalization

As it was mentioned in the Section 2.3.1, for improving the drug encapsulation efficiency of carriers, CCPs can be functionalized with additional LbL multilayers—the sequence of alternated adsorption of positive and negative polyelectrolytes [111,112,113].

The LbL multilayer assembly can effectively prevent the leakage of loaded molecules, thus keeping a high encapsulation efficiency and providing the functionality of sustained release of carriers as well as preventing water-induced phase transition and dissolution. Sergeeva et al. [114] demonstrated that the coating of CCP with additional layers increases the recrystallization time from 1–2 days till 20 days. These carriers maintained a good stability without changes of structure in the physiological environment at pH of 7.4, the release content was only 15% after 12 h of incubation, and the release amount increased with an increase of acidity, confirming the pH responsiveness of multilayers carriers.

In addition, these carriers could provide cell targeting by such molecules as folic acid (FA) [115], peptide (KALA) [116], therapeutic phosphorylated EEEEpYFELV (EV) peptide [117], PEG [118], hyaluronate/glutamate [119]. For example, Xing et al. [115] used chitosan and sodium alginate (SA) as coatings and modified the surface with PEG and targeting molecules FA to prepare multilayer DOX/CCPs carriers via the co-precipitation method combined with the LbL technology (Figure 6). In vitro experiments illustrated that DOX release rate of the hybrid system was slow, where only 35% of the total encapsulated amount of loaded drugs was released after 150 h of incubation, indicating that the system had long-term drug release characteristics caused by extensive electrostatic interactions between the polymer electrolyte and drug.

Another way of functionalization of particles is creating a silica layer, which would facilitate the stability of particles as well as the release rate of drugs. For example, when DOX/CCPs carriers were coated with a silica shell (DOX/CCPs@silica), the DOX in containers was desorbed firstly from DOX/CCPs via the phase transition and dissolution mechanism in the presence of water and acid, followed by the release of the free DOX into the external environment through the silica shell [101]. The release efficiency of the carrier was the highest at pH = 6.5 (25%) rather than that at pH = 5.5, and the amount of released DOX was extremely low at pH = 7.4 (<10%), because DOX (with pKa of 8.2) was positively charged at these pH values, while DOX/CCPs@silica nanoparticles were negatively charged between pH 5.0 and 7.4 [120,121]. The electrostatic attraction generated by these opposite charges was enhanced by the increase of the protonation degree of DOX induced by H^+^ [122], thereby weakening the release effect of DOX. Conversely, H^+^ also promoted the dissociation of DOX from DOX/CCPs core to accelerate the drug release. Therefore, the release mechanism of DOX was controlled by the synergistic effect of the above two factors, rather than due to a simple unidirectional increase as pH value decreased from 7.4 to 5.0. This is the reason why composite particles were different from single CCPs carriers, which had the effect of accelerating drug release with the decrease of pH.

Availability, productivity, mild decomposition conditions, and a favorable structure with a high surface area of CCPs make them a good candidate for sacrificial templates for the preparation of multilayer hollow capsules for targeted drug delivery and release [123]. In the review by Parakhonskiy et al., the authors summarized and analyzed possibilities of multilayers deposition of the LbL coatings (polyanion/polycation) on the surface of mesoporous vaterite, which, upon application of ethylenediaminetetraacetic acid (EDTA), dissolves producing polyelectrolyte multilayer capsules [124]. The structure of these capsules may be not completely hollow since the polymers penetrate into the crystal voids during the deposition of the inner layers, and when vaterite cores are dissolved, the polymer filling the voids may be retained within the capsules. These type of polyelectrolyte multilayer capsules showed feasibility of using CCPs as good templates. We note that Alg capsules prepared using CCPs as a template are summarized in Section 5. In the following section, we discuss alginate-based carriers.

## 3. Alginate-Based Hydrogel Carriers

### 3.1. Character of Alg-Based Hydrogel

Essential properties of Alg-Hs are porosity, swelling behavior, gel strength, immunological characteristic, biodegradability, and biocompatibility, which have a significant influence on their biomedical applications [125]. These physicochemical and biological properties are determined by the source, composition, and molecular weight of Alg as well as the gel-forming kinetics. Alg polymer is composed of two binding blocks, namely β-d-mannuronopyranosyl and β-l-guluronopyranosyl units. The monomeric units consist of β (1→4) linked β-d-mannuronic acid (M) residues and their C-5 epimer α-l-guluronic acid (G), which are referred to as M blocks (poly(mannuronic acid)), G-blocks (poly(guluronic acid)), or alternating regions MG-blocks [30,126]. The structure and properties of Alg are highly affected by the M/G ratio and the structure of attracting zones, which alters between different types of brown algae and even various pieces of the same plant. Alg-Hs enriched with poly G-block units exhibit a high porosity, low shrinkage during gelation, and do not swell after drying. On the contrary, Alg-Hs with plenty of poly M-block units becomes softer, more elastic with a reduced porosity and shrinking capability [127]. A high G-block content ensures that the resulting gels are more rigid, mechanically stable and less prone to erosion [128]. The gelation rate is an important factor in controlling the gelation process, which, in turn, affects the properties of gels. Slow gelation provides Alg-Hs a greater mechanical integrity and more structural uniformity [129].

### 3.2. Synthesis of Alg-Based Hydrogel

In view of the complex structure of organic molecular chains, more approaches can be employed to synthesize Alg-Hs compared to CCPs, mainly reflecting various methods for cross-linking (gelation process) of molecules or groups, and finally forming polymers with a low solubility and corresponding network structures [130]. Alg-Hs can be divided into two categories according to the crosslinking ways of network. One is ‘physical’ or ‘reversible’ gel depending on molecular entanglements, hydrophobic interactions and ionic or hydrogen bonding to keep networks together. And the other is ‘chemical’ or ‘permanent’ gel, which utilizes stable covalent bonds to form a network structure [131]. Overall, many approaches have been used to prepare Alg-Hs, including ionic crosslinking, covalent crosslinking, cell crosslinking, enzymatic crosslinking, phase transition and free radical polymerization [28,132].

Alg solution is easily converted into hydrogels by ionotropic gelation with polyvalent metal ions leading to the formation of a tough three-dimensional (3D) gel network, since the two G-blocks of adjacent chains can form electronegative cavities for carrying cations [133]. Ionotropic gelation can be further divided into external or internal mechanisms which differ with the source of crosslinking agent and gelling kinetics. External gelation adopts a soluble salt such as calcium chloride (CaCl_2_) to diffuse into the “sol” phase of Alg to form heterogeneous hydrogels. Internal gelation involves the incorporation of inactive agents and active crosslinkers to the “sol” phase of Alg to release cations through controlled alterations of the system properties such as pH or ion solubility to form homogeneous hydrogels (Figure 7A) [134,135,136]. During the process of gelation, each cation is bound with four G residues to form the typical “egg-box” structure (Figure 7A) [133,134,137]. The properties of ion such as the valence and radius substantially determine gelation kinetics and features of Alg hydrogels. Several divalent cations can be employed as crosslinking agents, and their affinity with Alg is graded as Mn^2+^ < Zn^2+^ = Ni^2+^ = Co^2+^ < Fe^2+^ < Ca^2+^ < Sr^2+^ < Ba^2+^ < Cd^2+^ < Cu^2+^ < Pb^2+^ [138]. Herein, some typical Alg-Hs prepared by ionic crosslinking are listed in Table 2, where CaCl_2_ is commonly used as an ionic crosslinking agent owing to its high solubility in an aqueous medium. But recent studies demonstrated that Ag ions could also be successfully used as cross-linking agents. Lengert et al. [139] demonstrated that using the template-based method for adsorbing alginate molecules, followed by cross-linking by Ag ions, could form silver alginate shell, which remains stable after the template dissolution. Following Ag reduction, the shell stability could decrease but provide additional functionalization by Ag nanoparticles. The advantages of ionic crosslinking are as follows: it is simple, high effective, environmental-friendly and can be carried out under mild conditions, while the major limitation is that the formed gels will disintegrate in a physiological environment [126].

Covalent crosslinking of Alg-Hs was initiated by crosslinkers through a copolymerization or polycondensation reaction between two polymer chains (Figure 7B). The functional groups, including –OH and –COOH, in Alg branches play a key role in the reactions with crosslinkers, such as glutaraldehyde, adipic acid dihydrazide, and poly(ethylene glycol)-diamine, and the mechanical properties of Alg-Hs with 3D networks formed via this way are mainly controlled by the crosslinking density and agent type [126]. Table 2 summarizes some researchers about preparing Alg-Hs through covalent crosslinking. Compared with ionic crosslinking, this approach exhibits more stable and controllable characteristics during the formation of Alg-Hs. However, the crosslinking agents tend to be toxic to the cells or tissues in vivo, so they need to be removed carefully and completely before applying hydrogels. In addition, the synergistic effect of ionic and covalent crosslinking mechanism provides feasibility for the synthesis of composite hydrogels with better mechanical properties, swelling ratio and antibacterial properties, and thus expands the application range of hydrogels [155].

Cell crosslinking strategy refers to the involvement of cells assisting the production of Alg-Hs networks through specific receptor-ligand interactions (Figure 7C). Despite Alg exhibits a good biocompatibility, it is composed of inert monomers and inherently lacks the bioactive ligands required for cell anchoring and adhesion. In the cell crosslinking method, first, ligands such as arginine, glycine, and arginine-glycine-aspartic acid (Arg-Gly-Asp, RGD) are introduced into Alg for cells adhesion by chemically coupling utilizing water-soluble carbodiimide chemistry. Then, homogeneous addition of mammalian cells to the ligand-modified Alg solution enables the receptors on the cell surface to crosslink with the ligands forming long-distance and reversible polymer networks even in the absence of chemical crosslinkers. Park et al. [156] prepared shear-reversibly crosslinked Alg-Hs by introducing both cell crosslinking and ionic crosslinking to Alg solutions, which efficiently regenerated new cartilage tissues in vivo. However, although cell crosslinking hydrogels show excellent bioactivities, and may be ideal carriers for cell delivery in tissue engineering, the low strength and toughness may limit their practical applications. Currently, the studies in this area are not extensive and further research is needed.

Enzymes can efficiently and specifically catalyze chemical reactions in a short time, while reducing the generation of by-products. Horseradish peroxidase (HRP) is a member of the large class of peroxidases with commercial availability due to their mild reaction outcomes, crosslinking efficiency, and excellent cytocompatibility. It has been reported that Alg can be catalyzed by HRP to form hydrogel networks in the presence of hydrogen peroxide (H_2_O_2_), in which the phenolic hydroxyl (Ph) groups were oxidized into polyphenols linked at the aromatic ring by C-C and C-O coupling between the Ph groups (Figure 7D). Liu et al. [157] coupled Alg and gelatin derivatives with Ph moieties (gelatin-Ph and Alg-Ph) first, and then collected Alg-Ph/gelatin-Ph hydrogels via HRP-catalyzed crosslinking. This conjugated hydrogel supported the growth of enclosed cells into spherical tissues and provided a cell adhesive outer surface for the fabrication of human aortic endothelial cell layer. Sakai et al. [158] prepared Alg-Ph hydrogel with different concentrations of HRP and H_2_O_2_, and investigated the effects on cellular adhesiveness and proliferation. Besides, incorporating Ph groups into polymers for enzymatic crosslinked gelation has also been proved to be effective in other biocompatible materials like hyaluronic acid, gelatin, carboxymethyl cellulose [159,160,161,162].

Thermo-responsive phase transition has been employed for the formation of hydrogel, because gelation can be realized simply as temperature increases above the lower critical solution temperature (LCST) [163]. Poly(N-isopropylacrylamide) (PNIPAAm) is one of the most extensively exploited thermo-sensitive agents, partially since the LCST in an aqueous solution is 32 °C. When the backbone of Alg was functionalized with PNIPAAm, a temperature-dependent behavior of the copolymer as well as enhanced mechanical strength and biocompatibility were achieved. Figure 7E shows the schematic representing the temperature dependent behavior of PNIPAAm grafted Alg hydrogels [28]. It is notable that PNIPAAm based Alg solution with great thermo-sensitivity could transform to hydrogel at physiological temperature. Tan et al. [164] synthesized a thermo-sensitive comb-like copolymer by grafting PNIPAAm–COOH with a single carboxy end group onto aminated Alg through amide bond linkages. The hydrogel was not cytotoxic and preserved the viability of human bone mesenchymal stem cells encapsulated in the copolymer well.

Free radical polymerization is the process of transforming linear polymers into 3D polymer networks with appropriate chemical initiators at the physiological pH and temperature. Methacrylated Alg with unsaturated C=C double bond groups is suitable for photocrosslinking to form covalent crosslinked bonds, which has recently received attention for various applications. Figure 7F represents the crosslinking process of photocrosslinking with methacrylated Alg. Many researchers have been interested in creating Alg-Hs by the free radical polymerization technique as delivery vehicles for biomedical applications (Table 3).

### 3.3. Drug Loading and Release

#### 3.3.1. Drug Loading

Drug loading must be taken into account when designing Alg-Hs carriers. The selection of preparation method, the amount of crosslinking agent or polymer and the species of bioactive agent can successfully control the size of hollow hydrogel, adjust the pore size, and modify the charge on the surface of hydrogel, thus improving the loading capacity of carriers. Alg-Hs can be loaded with drugs through physical adsorption and co-synthesis, and the significant differences in the physical properties of Alg-Hs and CCPs determine that Alg-Hs are more inclined to use the latter in terms of drug loading.

##### Physical Adsorption Loading Method

In physical adsorption process, the prepared Alg-Hs nano or microspheres are mixed with a solution containing drug molecules, while the interaction and adsorption between drug molecules and Alg-Hs take place through external forces such as stirring and vibration to complete the loading process and improve the loading efficiency. Similar to CCPs, the loading efficiency of Alg-Hs is also related to gel size, properties of drug molecules, drug solution concentration, mixing time, etc.

In general, larger Alg-Hs microspheres have a higher loading efficiency as they possess a wider internal space for carrying drugs, but the expansion of microspheres is also affected by the concentration of salt solution (NaCl) [172]. Additionally, over-dimension will lead to larger pore size, which will cause the leakage of drugs, and the delivery options for the drug-loaded system will be limited [173]. Furthermore, a higher concentration of drug molecules in the solution combined with the appropriate stirring time has a positive effect on improving the loading capacity until the maximum loading limit of the carriers (the equilibrium of adsorption-desorption kinetics) is reached [174].

At present, given the progress in the area of LbL multilayer coating, Alg-Hs can be coated with polyelectrolyte multilayers to significantly improve loading and encapsulation efficiency. Even a single polymeric layer and certainly multilayers can prevent the drug molecules originally adsorbed on the surface Alg-Hs from desorbing and leaking [175,176]. When Alg capsules are prepared based on the template method, the drug is adsorbed on the porous template through adsorption and cross-linking of Alg molecules. In this case, the loading efficiency is strongly dependent on the loading ability of the template and spontaneous diffusion during the deposition of gel molecules [75,139,177].

##### Co-Synthesis Loading Method

Similarly to the co-precipitation method, the drug loading process can be also accomplished during the synthesis of Alg-Hs via co-synthesis method. Generally, specific drug molecules and polymer substances used to improve the loading efficiency are added to Alg-Na solution, and then cross-linking of the polymers can be induced by nontoxic Ca^2+^ to form the drug carrier system [178,179]. In this case, the introduction of polymer substances and the increase of the concentration of Alg-Na solution will increase the viscosity of the mixed solution, so that more drug molecules will be trapped during the gel formation process [180,181,182]. However, a substantially high viscosity will result in overly large size of the final microspheres, thereby limiting the variety of delivery modes, which was reviewed by Uyen [180].

As a cross-linking agent, Ca^2+^ can change the gel hardening speed by adjusting the interaction between the polymer chain and the cross-linking medium, thereby changing the loading efficiency of Alg-Hs, and generally, an appropriate increase in the Ca^2+^ concentration can improve the loading capacity [180,181,183].

The surface charge properties of drug molecules will also influence the loading efficiency of Alg-Hs carriers. The drug molecules with a net positive charge will be easier bound to Alg-Hs due to the negative surface charge of Alg-Hs, and, conversely, particles with a net negative charge will be repelled. For example, the loading efficiency of Alg beads for whey protein is significantly higher at pH = 3 than at pH = 5 and 7, since the protein exhibited a net positive charge at pH = 3, making it tightly bound to the Alg beads due to the electrostatic attraction. While at pH = 5 and 7, the surface charge of whey protein was reversed (a net negative charge), making it repulsive to Alg, thereby reducing the loading efficiency [142]. In addition, the deprotonation of the carboxyl group of Alg and the charge loss caused by the change of pH value will also affect loading efficiency [184].

In addition, drugs can be loaded on the soluble or porous particle substrates, subsequently applying Alg layers on the surface of these substrates, and the loading of the drug molecules by Alg would be then achieved with a high loading efficiency [185,186,187].

#### 3.3.2. Release Mechanism

The drug release mechanism of the Alg-Hs hybrid system is dominated by the properties of the Alg matrix and polymer layers as well as the encapsulated drugs [142,188,189]. Alg-Hs can maintain integrity in an acidic environment rather than dissolve like CCPs, but they possess swelling-shrinking and degradable properties, which determines that the drug molecules are released through a synergistic mechanism of diffusion and degradation, while the desorption process also cannot be ignored [180,190].

When such drug-loading system is immersed in the medium, the drug molecules attached to the surface of the carrier are first released through desorption, which explains the early burst drug release of the carriers without polymer coating [142,188,191]. At longer times, Alg-Hs matrix absorbs solution and swells, which, in turn, causes an increase of pore sizes of gel networks, making it easier for the drug molecules with the original concentration difference inside and outside the gel to diffuse into the medium [184,192]. Meanwhile, the edges of the gel will also be dissolved or eroded by the medium. Over time, the degree of degradation of the gel increases, leading to a sustained release of the drug.

The above-mentioned cross-linking caused by high concentration of Ca^2+^ can form a denser and more stable gel network, thereby exhibiting a better sustained release [193]. At the same time, gel particles with a smaller size have a higher surface activity due to an increase in the specific surface area, which provides a larger surface for the gel particles. This facilitates diffusion and degradation, ultimately accelerating drug release [194,195]. Since Alg-Hs shrinks in an acidic environment, the pores of the gel network will be reduced preventing escape of drug molecules and thereby reducing the drug release rate [184].

In addition, Alg carriers or capsules can be functionalized with Ag nanoparticles, making them sensitive for remote release mechanisms such as ultrasound stimulated release [177] or laser induced release [145].

## 4. Application of CaCO_3_ and Alginate Carriers

The excellent physicochemical and biological properties of CCPs and Alg-Hs endow them with broad application prospects in the field of delivery carriers. In the past decade, numerous studies have manifested that these two types of carriers are satisfactory in terms of the loading and delivery of drugs, genes, peptides, and antibacterial molecules, especially for the targeted delivery and controlled release of anti-cancer drugs based on their pH sensitivity. Herein, promising applications of both carriers are overviewed.

### 4.1. Controlled Oral Drug Delivery

Different pH-responsive mechanisms of CCPs and Alg-Hs determine their respective advantages in drug delivery applications. Oral drug delivery is a superior method of administration due to its simplicity, convenience, and minimal pain. However, many challenges limit their efficient delivery, such as enzymatic degradation, hydrolysis, and low permeability into intestinal epithelial cells of the gastrointestinal (GI) tract. In this context, the pH responsiveness of CCPs turns into a weakness, since the rapid dissolution of pure CCPs and a burst release of molecules in an acidic environment make it difficult for the carriers to pass through the environment of stomach to pathological sites. Nevertheless, the assembly of effective coatings on the surface of CCPs can effectively alleviate this deficiency.

Liu et al. [196] synthesized HA-coated CaCO_3_ nanocarriers (HA/CCNPs) by adopting folic acid-modified HA incorporated with a simple and green inverse microemulsion co-precipitation method. Controlled trials revealed that no free insulin (INS) existed after incubation in strongly acidic simulated gastric fluid (SGF) for 30 min, and only 9.2% of the total loaded INS was retained from uncoated INS-CCNPs carrier, while the HA/CCNPs carriers still remained 68.7% INS after incubation in SGF for 90 min and maintained spherical morphology without degradation. Obviously, the presence of HA coating greatly enhanced the persistence of CCNPs. Co-culture with HT-29 cells showed that the drug-loaded system could internalize into cells and release INS. The blood glucose level was then reduced from 374 ± 13.1 mg/dL to 156 ± 16.5 mg/dL after administering INS-HA/CCNPs orally into diabetic rats for 3 h, which has prolonged duration of the drug efficacy in comparison to a subcutaneous injection of INS solution. Verma et al. [197] utilized Alg-Na and chitosan as raw materials to coat CCMPs via the LbL technique and loaded 4.3 ± 0.14% of Hepatitis B antigens. Non-coated CCMPs exhibited a burst and complete release in SGF, while the LbL/CCMPs turned into hollow capsules even though the core was dissolved, the presence of highly charged polyelectrolyte layers and the cross-linking effect hindered the rapid release of the drug molecules: there, only 20% was released in SGF for 2 h, followed by 40% for 8 h. Moreover, in vitro and in vivo intestinal uptake of LbL/CCMPs by mouse macrophages (J774) and intestinal epithelial cells (in vivo) was higher than that of uncoated CCMPs. A variety of drugs, such as peptides [104], vaccines [198], 5-fluorouracil [148], etc., have been tested in this coated-CCPs system as well, and some studies are shown in Table 4. It can be seen from the above discussion that polymers play an important role in the oral drug delivery system with CCPs as the carriers, and a reasonable combination of the above two carriers can provide a feasible route to synthesize carriers with improved performance.

Alg-Hs could be designed and modulated as controlled oral drug delivery carriers to protect drugs from hostile environment and satisfy the need for treating certain diseases (Figure 8A). The pH-sensitive composite gel beads composed of agar and Alg were crosslinked by Ca^2+^ with agar as a modifying agent [215]. The results illustrated that the properties of gel beads including mechanical strength, swelling and drug release behavior evaluated in simulated intestinal fluid (SIF) and SGF at 37 °C were affected by the content of agar. The beads with a higher agar content loaded with indomethacin (IDMC) exhibited a lower rate of swelling and drug release within 720 min. At pH = 2.1, the IDMC loaded beads shrank and hardly swelled attributed to the polymer interactions force originated from hydrogen bonds that dominate the polymer-water interaction force. The carboxylic groups transformed into –COO– at pH = 7.4 and hydrogen bonds between H_2_O and –COO– were formed leading to swollen particles (also referred to as beads), which release the drug [215]. Surya et al. [216] reported two chemical modification methods of Alg to provide neutral-basic pH sensitivity of the resultant hydrogel. In the first method, amide bond was formed directly between Alg and 4-(2-aminoethyl) benzoic acid. The second method involved the reductive amination of oxidized Alg to effectively control the degradation rate of the hydrogel. The hydrogel made by each method was capable to encapsulate payload and stay completely intact after 6 h in the harsh acidic pH conditions, but it gradually degraded within minutes to hours at neutral-basic pH conditions. Sevil et al. [217] developed INS entrapped alginate-gum tragacanth (ALG-GT) hydrogels at different ratios of Alg to gum tragacanth (1:0, 1:1, 1:3, 3:1) through an ionotropic gelation method, followed by chitosan polyelectrolyte complexation. INS efficiency was improved by the mild gelation process without any toxic chemicals. The hydrogel ensured that there was almost no leakage of INS in SGF and the maximum cumulative release up to 70% after 6 h in SIF. When the hydrogel moved from the gastric environment to the intestine, the negative charge of Alg and gum tragacanth was higher, sequestering the positive charge of chitosan, which allowed to the gel dissolution and INS release through electrostatic interaction [217]. Alg-Hs loaded with various drugs or biological compounds like ciprofloxacin [218], emodin [219], diclofenac sodium [220], bovine serum albumin [221], polyphenols [222], etc. have also been developed and investigated for controlled oral delivery system (Table 5).

### 4.2. Colon-Specific Drug Delivery

Colon specific delivery facilitates drug absorption and can be intended for the local treatment of diseases associated with gut, such as colon cancer, ulcerative colitis, irritable bowel syndrome, etc. After oral administration, the time required for the drug to reach colon is expected to be around 8 h. Therefore, an ideal colon specific delivery system should delay release of the drug in the upper GI tract, stomach, and small intestine with a long lag time and guarantee the maximum release amount in the lower GI tract (large intestine or colon) (Figure 8B). CCPs carriers have also been applied to colon specific drug delivery. Chen et al. [244] employed water-in-oil (W/O) emulsion to synthesize CCPs encapsulated liposomes (LCC) with dimension of 155 nm to deliver curcumin. Although the pH sensitivity of CCPs makes them vulnerable to acidic environment, which would affect the stability of lipid bilayers, the LCC maintained structural integrity at pH = 5.5, and the release of curcumin from LCC occurred continuously, without a burst release. Compared with free lipophilic curcumin and curcumin/liposome (CLIPO) carrier, both of which were no efficiently taken up by cells, whereas LCC was ingested significantly in cell culture medium for 4 h. In vitro cytotoxicity results illustrated that the half maximal inhibitory concentrations (IC50) of free curcumin, CLIPO and LCC were 9.413, 7.166 and 5.067 μM, respectively, indicating that LCC enhanced the binding of curcumin to cells through effective lysosomal escape and drug release, which in turn promoted the efficient inhibitory effect of proliferation. Moreover, LCC exhibited the best antitumor effect in the azoxymethane (AOM)/dextran sodium sulfate (DSS)-induced colorectal cancer model, since long circulation time and efficient intracellular delivery of curcumin from LCC improved pharmacodynamics of curcumin. Some additional studies are shown in Table 4. Likewise, Atitaya et al. [245] developed Alg beads containing self-emulsifying curcumin (SE-Cur) by ionotropic gelation and coated with Eudragit^®^ S-100. The encapsulation efficiency of poorly water-soluble curcumin was in the range of 85–98%. The results indicated that 2–4% of Alg and 0.1 or 0.3 M of CaCl_2_ could prevent curcumin release in the upper GI tract and >60% of the drug was released upon the arrival of the beads in simulated colonic fluid within 12 h. In addition, SE-Cur released from the beads showed cytotoxic ability against the human colon adenocarcinoma cell lines with 10 μg mL^−1^ of IC50 and potential antioxidant activity assessed by the reducing power assay of ferric cyanide (Fe^3+^). Sheng et al. [224] designed a novel dual-drug delivery system based on the Alg and sodium carboxymethyl cellulose (CMC) hydrogels for colorectal cancer treatment. In this system, methotrexate (MTX) loaded CaCO_3_ (CaCO_3_/MTX) was first prepared via co-precipitation method, and then was co-entrapped with aspirin (Asp) in the hydrogels of Alg and CMC crosslinked with Ca^2+^. The results of MTX and Asp delivery revealed that hydrogels can prevent the release of two drugs during the first 2 h in SGF, and the cumulative release of Asp increased significantly up to 57.2% from 2 h to 8 h in the SIF, while the release of MTX was still suppressed with only 12.8% during this period. When the ambient pH was the same as that in the colorectum (pH 6.5), the release of MTX was greatly enhanced to 47.6% from 8 h to the 24 h; this is because CCMPs was gradually decomposed at pH 6.5 and the hydrogels maintained a relatively high swelling ratio of 32.4. The dual-drug delivery system also showed the cytotoxicity on the SW480 colon cancer cells with a prominent concentration dependence, and the viability of the cancer cells was declined to less than 10% at the concentration of 128 μg·mL^−1^. Lengert et al. [145] demonstrated that Alg-Ag gel with Ag nanoparticles successfully penetrated intestine of worms and was capable of releasing the payload under near-infrared laser irradiation. Additional data on the colon targeting delivery systems based on Alg hydrogels are listed in Table 5.

### 4.3. Transdermal Drug Delivery

Non-invasive transdermal administration is an attractive route due to its low proteolytic activity and sustained delivery and absorption of drugs, but the disadvantage is that the skin penetration rate is rather low [246]. Relatively good mechanical properties of CCPs allow them to penetrate the superficial skin smoothly and deliver the carried drugs to deeper tissues without being damaged, which can be facilitated by external stimuli such as ultrasound, and finally complete the drug release through sustained release mechanism and pH sensitivity [205,206].

In this regard, the preclinical performance of uncoated naftifine (NfCs)-loaded CaCO_3_ carriers with 4.9 wt% of loading capacity were evaluated by Gusliakova et al. [247]. In contrast to free drug solution, the NfCs system had twice the fungal growth inhibition rate, and the fluorescence analysis after co-culture of carriers and cells proved that the carriers were highly taken up by cells and had an extremely low cytotoxicity. A NfCs suspension was applied to the back of mice and ultrasound was performed to drive drug delivery. SEM images of hair root sheath removed from mouse skin in separate periods showed that NfCs penetrated successfully into the hair follicles, followed by gradual degradation inside the hair follicles within 120 h. The majority of NfCs were absorbed within 72 h, ensuring the release of the payload into the tissue surrounding the hair follicle. Thus, the presence of porous CCPs allows delivery of naftifine hydrochloride loaded into the matrix to the deeper layers of the skin, providing an enhanced intradermal accumulation. Moreover, NfCs coated with heparin, poly-l-arginine, and dextran sulfate (DS) layers endowed uncoated NfCs with longer payload release time (>72 h), so as to play a sustained release effect and long-term efficacy [247]. Diabetes mellitus is a group of metabolic diseases characterized by hyperglycemia owning to defective INS secretion or impaired biological action. The therapeutic drug INS is highly degraded in the digestive tract and, given its low skin permeability as well as the risks of injectable administration, transdermal drug delivery opens new prospects for drug delivery. In this regard, CCPs and CCPs-based hybrid carrier not only possesses impressive drug-loading capacity and pH sensitivity, but also endows the carrier strong mechanical strength, so as to maintain the integrity of the carrier and smoothly pass through the skin tissue. For instance, Liu et al. [248] selected INS-loaded CaCO_3_ microparticles (INS-CaCO_3_ MPs) and polyvinylpyrrolidone (PVP) as matrix to fabricate dissolving microneedles (INS-CaCO_3_/PVP MNs), as shown in Figure 9. The hybrid carriers exhibited a good strength and toughness compared to the PVP/MN carriers and were capable of penetrating the skin. In the experiments performed on the back of rats, morphological retention time of INS-CaCO_3_/PVP MNs was prolonged, resulting in a longer release time of INS and sustained release performance. The discovery of FITC-labeled INS at deeper tissues manifested the strong skin insertion ability of the carriers, and the in vivo pharmacokinetic study of diabetic mice showed that INS-CaCO_3_/PVP MNs had a more efficient and sustained INS release capability than that upon a subcutaneous injection, and the relative pharmacological availability and relative bioavailability of the carrier to INS were 98.2 and 96.6%. Additional studies are reported in Table 4.

Drug administration through the skin is performed in order to treat skin diseases topically or for the transdermal absorption of drugs into the systemic circulation. Hydrogels plays an important role in skin elasticity, moisturization and repair because they are not only capable of holding substantial amount of water, but also control the release of loaded drugs. This makes them suitable candidate for topical use (Figure 8C) [225]. For example, Ludovico et al. [249] designed and characterized a novel series of hybrid hydrogels based on Alg and poly(vinyl) alcohol (PVA) for enhanced skin delivery of quercetin. The rheological analysis of quercetin-loaded hydrogels displayed pseudoplastic behavior, non-thixotropy and good resistance to deformation in the range of 20 °C to 40 °C. The hybrid hydrogel retained antioxidant activity of quercetin and facilitate its entry in the skin. These therapeutic hydrogels represent a potential topical transdermal drug delivery system for the treatment of skin ageing and inflammation. In other studies, Li et al. [250] fabricated a bioinspired Alg-gum Arabic hydrogel with micro/nanoscale structures to deliver mitsugumin 53 (MG53), an essential component in cell membrane repair, for chronic wound healing. In vitro experiments showed that the hydrogel had a biphasic-kinetics release, which can facilitate both fast delivery of MG53 for improving the reepithelization process of the wounds as well as sustained release of the protein for treating the remodeled phase of chronic wounds. In vivo mouse model experiments confirmed that the hydrogel encapsulated within rhMG53 could promote dermal wound healing and suppress scarring without adverse inflammation and abnormality of major organs. Zohreh et al. [251] packed hesperidin in Alg/chitosan hydrogel as an efficient method for wound healing. The hydrogels possess appropriate porosity (91.2 ± 5.33%) with the interconnected pores and could degrade almost 80% after 14 days. In vitro results revealed the hydrogels have no toxicity on 3T3 murine fibroblast cells and the negative effect on *Staphylococcus aureus* and *Pseudomonas aeruginosa*. The therapeutic efficacy of the hydrogel was confirmed on the healing of skin injury in a rat model. The hydrogels loaded with 10% of hesperidin had highest wound closure percentage than other different concentrations of non-encapsulated hesperidin groups. Table 5 summarized additional examples of Alg-Hs used for transdermal delivery drug.

### 4.4. Ocular Drug Delivery

Eye drops are the most prevalent administered ocular medications, but they are difficult to retain in the tear film for long periods and have insufficient contact with epithelial cells [252]. Previously, superoxide dismutase (SOD), a drug for the treatment of various eye diseases related to oxidative stress, has been extensively studied in combination with various organic carriers [253]. However, SOD is easily inactivated by protease hydrolysis and is rapidly cleared, so it is necessary to improve the topical administration and bioavailability of SOD [254]. Nanoscale CaCO_3_ carriers with adhesion and a series of excellent properties are more attractive, although they have not yet been widely studied in ocular drug delivery related articles. Some examples include the following: Binevski et al. [253] injected vaterite microcrystals at a concentration of 10 mg/mL into the eyes of rabbits and did not find any negative reactions. On this basis, the loading capacity, release mechanism and related physicochemical properties of SOD/CCNPs carriers were investigated. The size of SOD molecules was estimated to be 6 nm in TRIS buffer; molecules with such sizes penetrated into the pores of vaterite (5–30 nm) or were embedded in the particles during co-precipitation without any hindrance, and the loading capacity and encapsulation efficiency can be improved up to 20% and 90%. The release of SOD/CCNPs in NaCl solution reached 80% in 100 min, and the SOD activity retention was as high as 85%. After 24 h and 36 h, the activity retention of the enzyme decreased to 50% and 30%, respectively, since long-term exposure to the physiological ion solution would reduce the activity of the enzyme [253].

Efficient ocular drug delivery remains a challenge for ocular diseases (e.g., conjunctivitis, age-related macular degeneration (AMD), glaucoma and diabetic retinopathy), due to the physiological barriers of the eye and the complexity and particularity of the anatomy. Hydrogels as innovative delivery methods have been explored for ophthalmic medicine applications and introduce possible solutions to overcome the ocular environment. An in situ injectable glycol chitosan and oxidized Alg cross-linked hydrogel was synthesized to encapsulate Avastin^®^ for potential ocular drug delivery to treat AMD (Figure 8D). In vitro release study demonstrated that the entrapped Avastin had an initial burst release from hydrogels at early stage (within 4 h). After that, Avastin retained sustained release manner in period of 3 days without apparent structure changes. The release rate of Avastin was declined with the increase of oxidized Alg concentration [226]. Besides, terminalia arjuna gum/Alg, a pH triggered in situ gel system with prolonged retention time of moxifloxacin HCl (MOX-HCl) for ocular drug delivery was prepared and characterized. Physicochemical characterization and in vitro drug release profile of optimized formulation showed that the gel was stable, non-irritant, therapeutically efficacious and displayed sustained release for 12 h. Ocular irritation study via modified Draize eye test of rabbit indicated that instillation of the gel did not cause any type of irritation symptoms like redness, inflammation and excessive tear production [255].

### 4.5. Intravenous Drug Delivery

Intravenous administration is considered to be one of the most common and fast routes to provide systemic drug delivery when patients are unable to actively ingest drugs through mouth and nose (Figure 8E), for example, due to the loss of self-consciousness or other reasons. However, during circulation in blood, the targeting and efficacy of the drug will be reduced. Therefore, it is very important to select a carrier with good stability and targeting.

Lin et al. [256] studied the effects of CCPs on blood components (erythrocytes, platelets) and coagulation function by intravenous injection and found that they did not cause abnormalities in key tissue structures. This provided support for the subsequent application of CCPs in biomedicine. To achieve desired intravenous circulation and reduce the pain of injection, the particle size of CCPs serving as drug carriers generally does not exceed 120 to 200 nm. For example, DOX (an immunogenic cell death (ICD) inducer) in a combination with alkylated NLG919 (aNLG919) (an inhibitor of indoleamine 2,3-dioxygenase 1) were loaded into CCPs, and they improved the penetration of the drug into the tumor, exhibited efficient tumor aggregation, and neutralized the acidity of the tumor, resulting in an efficient drug chemotherapy [207]. Generally, degradable lipids or polymers, such as PEG, PAA are used to coat the surface of drugs molecular to enhance the stability of delivery systems [90,97,118]. In addition, the hybrid carriers can be doped with some functional metal nanoparticles to improve comprehensive performance. For example, magnetic nanoparticles like Fe_3_O_4_ have been used to improve the active targeting of drug carriers [257]. MnO_2_@CaCO_3_ or Fe@CaCO_3_ carriers formed by doping with manganese dioxide or iron showed good photodynamic and chemical kinetics therapeutic effect after loading such drugs as DOX, siRNA, oxidized cisplatin prodrug, polydopamine, etc. [68,97,227,258], since the loaded metal ions can catalyze the formation of oxygen from H_2_O_2_ to improve the hypoxic microenvironment at the tumor forming toxic hydroxyl groups to kill cancer cells. Additional studies are shown in Table 4.

Unlike CCPs, hydrogels can be elongated or stretched under an external force to pass through narrower channels without clogging them, since they possess high plasticity determined by their softness [259]. Therefore, for intravenous injection an Alg-based drug delivery system can have a larger size than that of CCPs carriers; in this regard, some typical particles with sizes in the range of 180–400 nm have been investigated exhibiting good capabilities for carrying drugs [260,261,262]. However, that does not rule out a preference for smaller-sized Alg particles [263,264,265]. In addition, the encapsulation of some drugs with a high hydrophobicity and poor bioavailability by hydrophilic alginic acid can improve the permeability and bioavailability of drugs [259,260,263,265]. For example, Saralkar et al. [263] prepared Ca-Alg nanoparticles with an average size of 50 nm loaded with hydrophobic curcumin and resveratrol via emulsification cross-linking method. The results showed that curcumin was effectively taken up by cells and had better expected effects than free drug. Metal-functional nanoparticles have been also introduced into Alg-Hs to improve the target responsiveness of hybrid carriers [266]. Additional studies are shown in Table 5.

### 4.6. Intranasal Drug Delivery

Given the limited accessibility of the brain and central nervous system, many traditional drug delivery methods are not capable of achieving efficient drug delivery of certain psychotropic drugs. Therefore, intranasal administration is considered to be an important drug delivery methods for brain targeting since the nasal passage is directly connected with the brain and it establishes the channels through which the brain interacts with the outside world (Figure 8F).

An INS drug delivery system based on intranasally administered CCPs achieved faster absorption and quicker reached a higher blood sugar level [267]. Compared with square CCPs, porous spherical CCPs had higher adhesion and drug delivery efficiency. The addition of polyelectrolyte multilayers can improve the adhesion of the carriers while achieving higher encapsulation efficiency, and finally achieve a higher nasal delivery effect. For example, Marchenko et al. [268] adsorbed polyelectrolyte multilayers on mesoporous CCPs carriers loaded with imidazopyridines (anxiolytic hypnotics) and visualized the carriers after intranasal administration using X-ray microtomography. Tomography images of the cross-section of nasal turbinate showed that a certain amount of formulation was remained after 1 h of administration. These results indicated that the hybrid carriers based on mesoporous CCPs had a longer half-life (15–30 min) in the nasal cavity compared with mucociliary clearance, which removes extraneous objects from the nasal epithelium, providing the feasibility of these hybrid carriers for nasal administration. Additional data are shown in Table 4.

Compared with CCPs which have a stable bonding structure, the network structure and existing functional groups of Alg-Hs make them easier to contact with the nasal mucosa tightly, thereby prolonging the retention time of the drug carriers and enabling them to be better absorbed by the nasal epithelial cells. Some psychotropic and neurological drugs have been encapsulated in Alg-Hs particles for intranasal administration, such as venlafaxine and albiflorin, which are used to cure depression. Alg-Hs particles adhere well to the nasal mucosa for continuous release and rapidly enter to the brain in short time, resulting in good antidepressant efficacy [228,269]. In addition, systemic vaccines do not usually produce mucosal immunity, which allows virus to break through the body’s first line of defense, Alg-Hs combined with intranasal inoculation were shown to effectively induce mucosal-specific immunity [270]. Additional studies are presented in Table 5.

## 5. Comparison and Combination of CaCO_3_ and Alginate

Applications of CaCO_3_ and Alg and their hybrids are compared and summarized in Table 6. We performed the comparison in terms of several parameters, including the methods of their preparation. Both types of containers are prepared via green synthetic efficient methods based on simple salts or biopolymer mixtures via kitchenware technology. The synthesis of CaCO_3_ is less controllable, and it mostly depends on the mass of the effect of crystallization [42,44]. The synthesis of Alg is more controllable but the size and shape of the objects depend on the facilities, such as the capillary for flow of hydrogel generation, and templates for their synthesis [131,132]. The size and shapes of CaCO_3_ particles are predetermined by the spontaneous synthesis of thermodynamically stable particles and are limited due to external conditions [42,44]. In turn, the fabrication of Alg particles depends on the template, enabling the synthesis of millimeter-sized capsules suited for oral delivery. CaCO_3_ is a hard mineral, whereas the Alg-based is a relatively soft material. The loading capacity of CaCO_3_ is determined by the porous structure of the mineral matrix and the molecules are adsorbed by the porous structure via electrostatic or Van der Waals forces. On the other hand, the loading capacity of Alg depends on the solubility of the drug in hydrogels. These two containers have different release profiles, and the more important factor influencing the drug release from CaCO_3_ is pH, while for Alg-Hs it is spontaneous diffusion of encapsulated molecules. Therefore, an appropriate combination of these two materials will overcome the shortcomings of a single material and absorb their respective advantages, thus showing improved performance compared to a single carrier.

Interestingly, the combination of these two excellent biomaterials has also been extensively studied (Figure 10). As mentioned above, the polyelectrolyte layer on the surface of CCPs can effectively improve the drug encapsulation and delivery efficiency, thereby achieving higher drug efficacy [115,281] (Figure 10A).

The CCPs loaded with drug molecules are placed in an Alg solution, and Alg-Hs layers can be assembled on the particle surface by appropriate selection of the method of inducing the cross-linking of the hydrogel. The formed hybrid drug-loaded system can effectively prevent the leakage of the encapsulated molecules and has a sustained release effect, because Alg can still be retained even after the dissolution of the CaCO_3_ core [177]. For example, Zhao et al. [272] loaded the antitumor drug DOX and DNA plasmids into Alg/CaCO_3_ nanoparticles (Alg/CaCO_3_/DOX/DNA) with a high encapsulation efficiency, and the carriers exhibited a high cell inhibition rate of about 80%, indicating that they could efficiently mediate gene transfection and deliver drugs to cells. Compared with CaCO_3_/DNA/DOX nanoparticles without alginate modification, the former showed a higher delivery efficiency. In addition, these hybrid particles can transform into porous or hollow Alg capsules by acid etching of CaCO_3_ core, but the soft properties and porosity of the capsules may limit their application (Figure 10A). Sergeeva et al. [282] prepared porous Alg gels by a one-step template method in Alg solution with sacrificial CCPs. The density of gel networks determined by the concentration of Alg solution and the dissolution rate of the CaCO_3_ core had important effects on the formation of pores in capsules. The amount of Ca^2+^ produced by a low dissolution rate of CCPs was insufficient to locally associate with Alg molecules for forming gel networks, which, in turn, made the gels unstable. Furthermore, the glucose molecules loaded into the template in advance will not affect the pore size of the gels due to the change of osmotic pressure during the core dissolution process. An extremely fast dissolution of CaCO_3_ core will instantly generate numerous CO_2_ bubbles, thereby expanding the pores of the gels, but this process is difficult to be well controlled. The accessible soft and hollow Alg-Hs have an important application in the fields of cell scaffolds and drug delivery systems. Importantly, such hollow Alg scaffold, which relies on the soluble CaCO_3_ core, has important application prospects in tissue engineering, which was reviewed in [283]. Furthermore, embedding CCPs into Alg-Hs can significantly enhance the mechanical properties of gels. In this case, the hard mineral particles incorporated in soft gel function as an adhesion center for cells [284] (Figure 10B,C). Additional studies are given in Table 7.

Overall, pure CCPs, Alg-Hs, or hybrid carriers composed of both, or Alg-Hs capsules obtained with CCPs as templates, have important research values and application prospects in many fields, especially in the drug delivery and tissue engineering.

## 6. Conclusions

Although CaCO_3_ and Alg-Hs belong to two distinct material types, namely inorganic and organic, their respective properties indicate that they are attractive either individually or through a carrier combining these particles for encapsulation and targeted delivery of therapeutic drug molecules as well as tissue engineering. Peculiarly, carriers based on these materials have similarities. Indeed, both types of materials are synthesized in an easy way using green and kitchen-based chemistry and they both were used in numerous biomedical and pharmaceutical applications. In addition, both types of materials can be coated with, for example, polymeric coatings or multilayers to control stability, release rate, surface functionalization, mechanical properties, etc. Naturally, there are differences between them.

Being hard, highly porous, and pH-dependent, CaCO_3_ allows for effective loading and targeted release of drugs. Furthermore, CaCO_3_ carriers are denser and are affected by an aqueous solution (water soluble), and it is more challenging to control their properties upon synthesis. On the other hand, soft and versatile Alg-Hs do have their own advantages: they are pH-stable, have a high-water content, enhanced elasticity and stretchability, possess stability in an aqueous solution (water stable).

However, there are still some challenges before these carriers can be used more widely. For example, although CaCO_3_ can be synthesized through facile methods, the expected nanometer size particles with a high porosity and good morphology are often unstable due to the phase transition. Ultimately, this will limit the long-term efficacy of this drug carrier. In addition, it is important to develop targeted drug delivery carriers based on CaCO_3_, which endows them with multifunctional properties. In contrast, Alg-Hs particles and gels suffer from an uneven ion gelation and relatively low mechanical strength compared to inorganic particles, which affects their application in biological sciences. In recent years, research on Alg-Hs has focused on enhancing their mechanical properties and increasing the range of biological applications. For example, thermal annealing has been employed to enhance the mechanical strength of Alg-Hs for tissue engineering [287]. Further, divalent strontium cations are introduced into Alg-Hs to promote excellent biological applications in osteogenesis, angiogenesis, and cell adhesion and extension. Combining Alg-Hs with other biomolecules to prepare new hybrid materials is viewed essential for improving the mechanical strength and versatility of Alg-Hs carriers.

Whether it is optimization of preparation methods for CaCO_3_ and/or Alg-Hs particles alone or formation of composite particles by involving both of these materials (inorganic-organic hybridization or hard-*and*-soft approach)-such approaches can be used to solve the above-mentioned challenges and develop versatile drug delivery carriers. In fact, combining hard inorganic CCPs with soft Alg-Hs to produce hybrid materials has already shown to be a good way to prepare drug delivery systems with favorable mechanical properties, high loading efficiency, sustained and controlled drug release capability. Moreover, such hybrid structures have led to improving performance of coatings, which allow to effectively grow cells on their surface [287]. Therefore, utilizing a high strength of inorganic CCPs combined with a good biocompatibility and plasticity of Alg-Hs is envisioned to confer the superiorities of these two distinct substances into one hybrid material, which provides a promising approach for drug delivery systems.

## Figures and Tables

**Figure 1 pharmaceutics-14-00909-f001:**
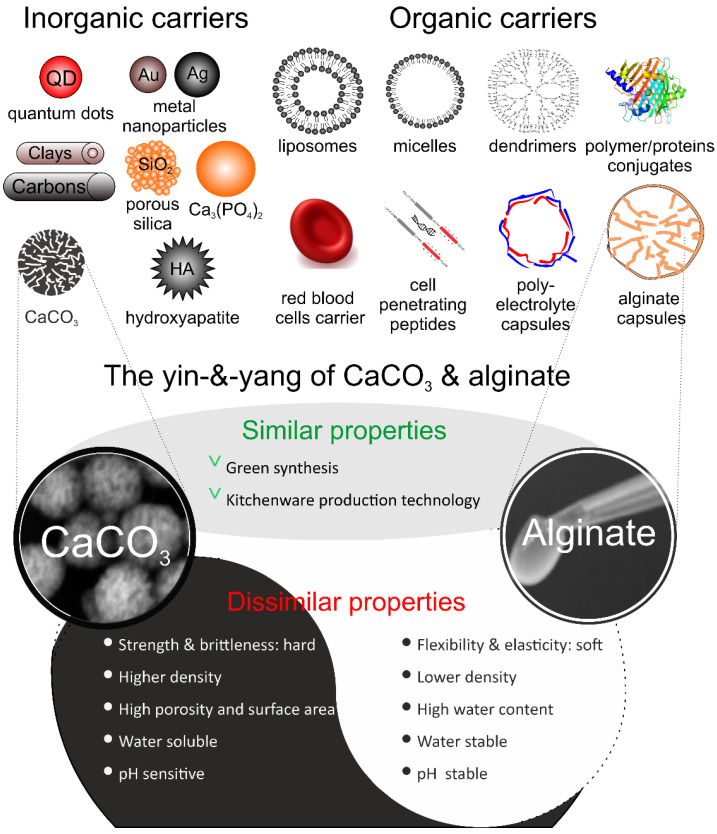
An overview of inorganic and organic drug delivery carriers including calcium carbonate and alginate carriers (**above**). Similarities and differences (yin-&-yang properties) between CaCO_3_ and alginate and their respective attributes (**below**).

**Figure 2 pharmaceutics-14-00909-f002:**
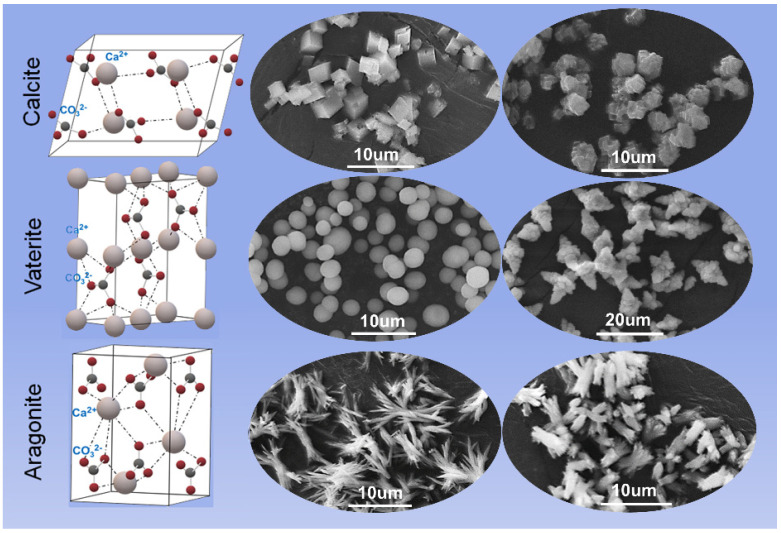
The crystal structure and macroscopic morphology (obtained from a scanning electron microscope (SEM)) of three crystalline phases of CaCO_3_ particles: calcite (**top**), vaterite (**middle**), and aragonite (**bottom**).

**Figure 3 pharmaceutics-14-00909-f003:**
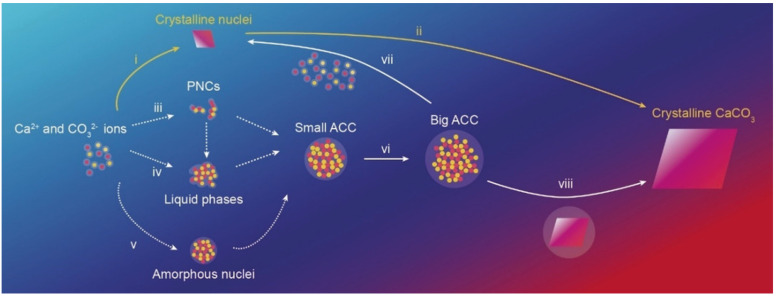
Schematic illustration of proposed mechanisms of the CaCO_3_ formation. Reproduced with permission from [53], John Wiley & Sons—Books, 2020. (PNCs: prenucleation clusters; ACC: amorphous CaCO_3_).

**Figure 4 pharmaceutics-14-00909-f004:**
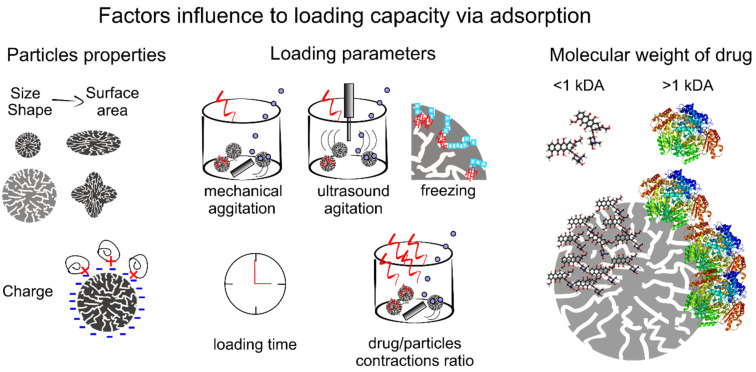
Factors influencing the loading capacity of CaCO_3_ particles via physical adsorption including particle properties (size, surface area, charge, shape), loading parameters (mechanical and ultrasound agitation, freezing, loading time, concentration), molecular weight of drugs (under 1 kDa and over 1 kDa).

**Figure 5 pharmaceutics-14-00909-f005:**
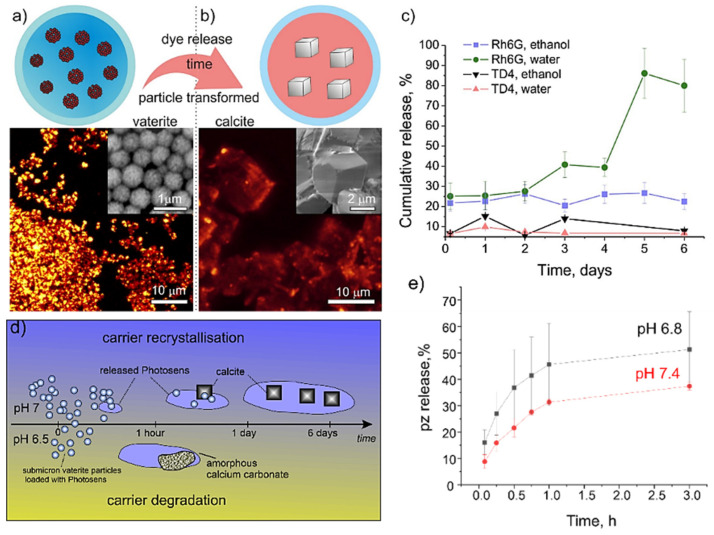
Capsule recrystallization in a medium. Panels (**a**,**b**) represent a scheme of the release mechanism and the corresponding container two-photon microscope fluorescence images with scanning electron microscopy (SEM) images as insets. (**a**) The carriers are in a pure vaterite phase with all dye encapsulated; panel (**b**) shows the calcite phase, where the dye was released into the medium apart from residuals attached to the crystal surfaces. Panel (**c**) displays dye release curves for different payloads measured by spectrofluorometry during the immersion in water and in ethanol. Panels (**a**–**c**) are reproduced with permission from [70], the Royal Society of Chemistry, 2013; panel (**d**) schematics of the degradation of particles due to pH changes. Reproduced with permission from [73], Elsevier, 2013. Panel (**e**) depicts release of the perphenazine. Reproduced with permission from [74], Elsevier, 2021.

**Figure 6 pharmaceutics-14-00909-f006:**
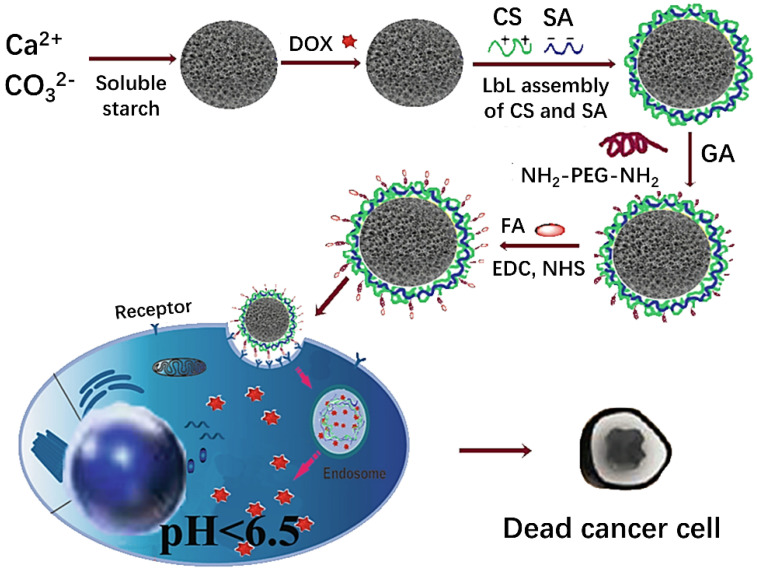
Coated CaCO_3_ nanoparticles were prepared using the LbL method for the targeted delivery and controlled release of drug molecules. Reproduced with permission from [115], Hindawi (open access), 2020. (DOX: doxorubicin; CS: chitosan; SA: sodium alginate; FA: folic acid; GA: glutaraldehyde; PEG: poly(ethylene glycol); EDC: N-(3-dimethylaminopropyl)-N′-ethylcarbodiimide; NHS: N-hydroxysuccinimide).

**Figure 7 pharmaceutics-14-00909-f007:**
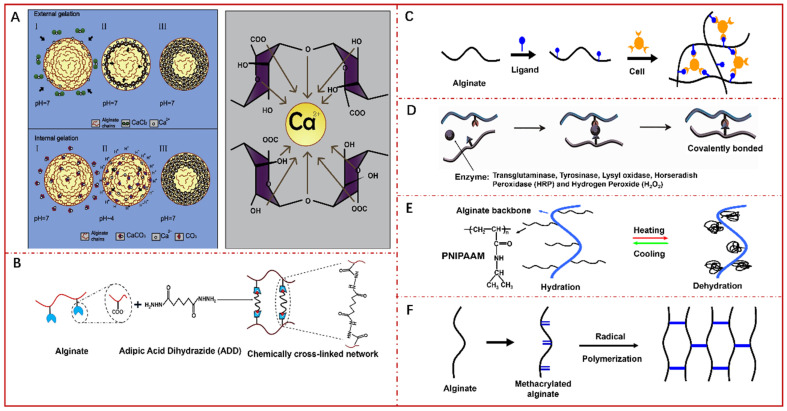
Schematic showing the gelation mechanism of alginate prepared using different methods. (**A**) Ionic crosslinking (Left); egg-box model of alginate gelation (Right). Reproduced with permission from [134], Elsevier, 2020; (**B**) covalent crosslinking; (**C**) cell crosslinking; (**D**) enzymatic crosslinking; (**E**) phase transition; (**F**) free radical polymerization. Reproduced with permission from [28,30], MDPI, 2013; Springer Nature, 2020.

**Figure 8 pharmaceutics-14-00909-f008:**
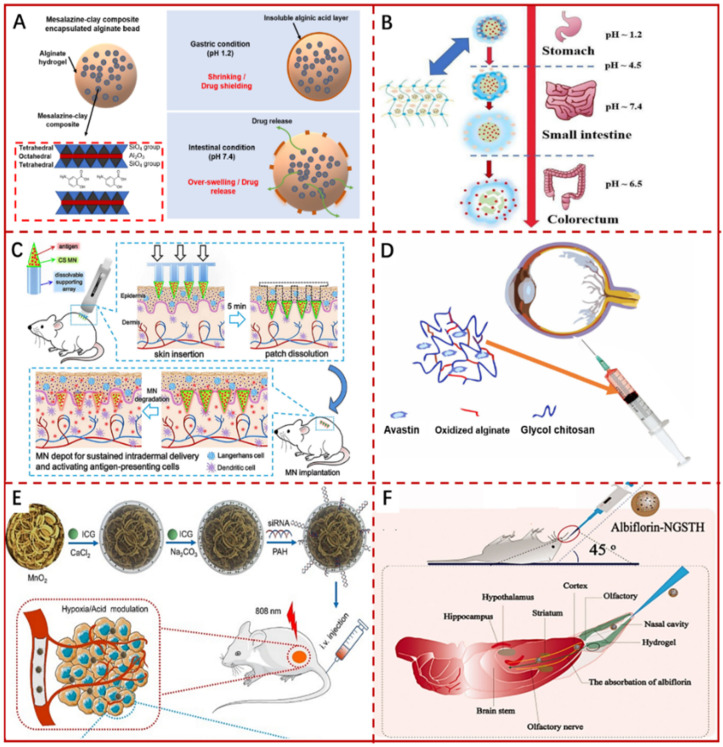
Applications of CaCO_3_ or alginate-based hydrogel carriers in drug delivery. (**A**) Oral drug delivery. Reproduced with permission from [223], Springer Nature, 2017; (**B**) colon-specific delivery. Reproduced with permission from [224], Elsevier, 2021; (**C**) transdermal drug delivery. Reproduced with permission from [225]. Elsevier, 2021; (**D**) ocular drug delivery. Reproduced with permission from [226]. Elsevier, 2013; (**E**) intravenous drug delivery. Reproduced with permission from [227]. Ivyspring International Publisher. 2019; (**F**) intranasal drug delivery. Reproduced with permission from [228], Taylor & Francis, 2021.

**Figure 9 pharmaceutics-14-00909-f009:**
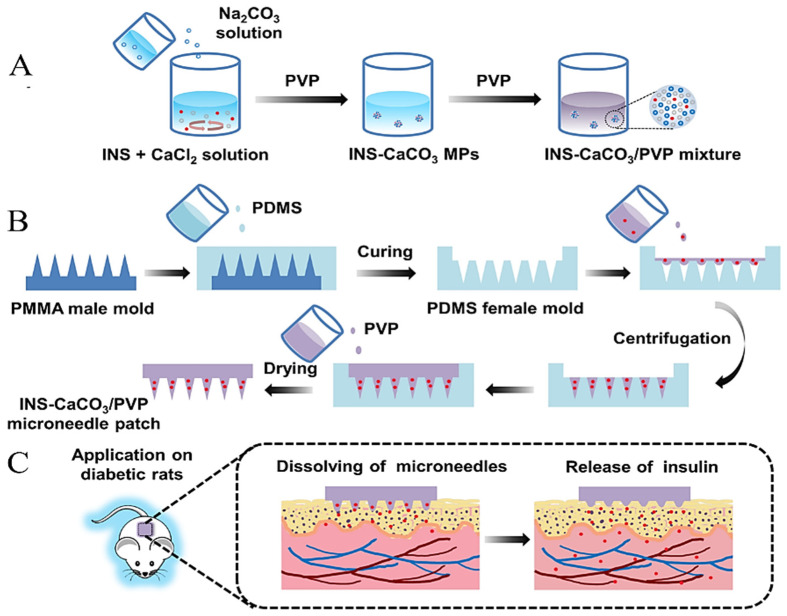
Schematic illustration of (**A**) preparation of INS-CaCO_3_ MPs through precipitation method in the presence of PVP as surfactant; (**B**) fabrication process of INS-CaCO_3_/PVP MNs from PMMA male mold, and (**C**) application of INS-CaCO_3_/PVP MNs on diabetic rats for transdermal delivery of insulin. Reproduced with permission from [248], Elsevier, 2018. (INS: insulin; MPs: microparticles; PVP: polyvinylpyrrolidone; MNs: microneedles; PDMS: Poly(dimethylsiloxane); PMMA: polymethyl methacrylate).

**Figure 10 pharmaceutics-14-00909-f010:**
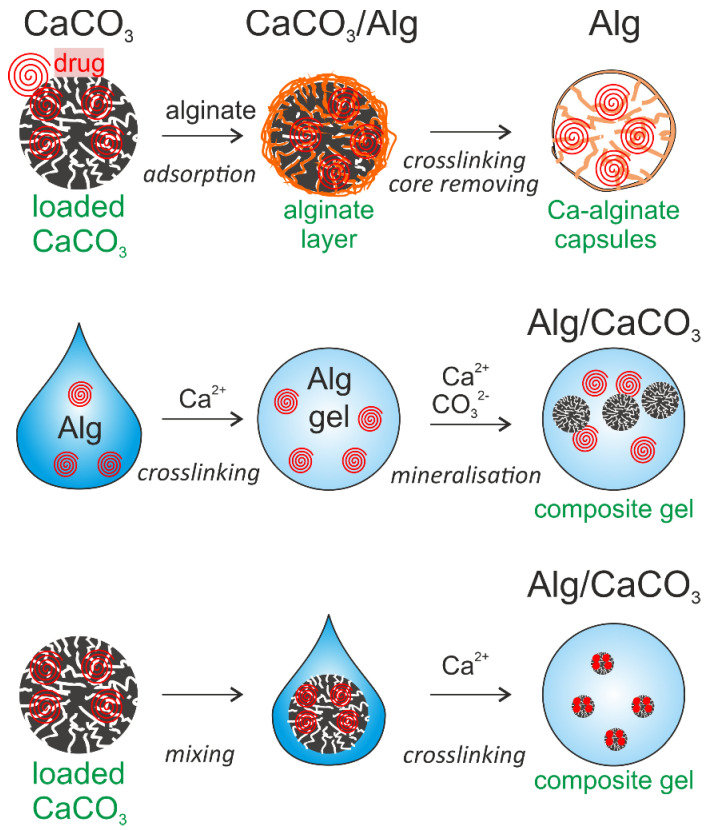
Various possibilities of combining CaCO_3_ particles and Alg, including (**A**) using CCPs as a template; (**B**) crosslinking Alg gels by a mineralization process; and (**C**) crosslinking Alg gels into a composite gel.

**Table 1 pharmaceutics-14-00909-t001:** Studies on the application of CaCO_3_-based carriers in drug delivery.

Particles	Loading Method	Particle Size	Loaded Drug	Loading Efficiency	Release Mechanism and Major Effect
CaCO_3_	Physical adsorption	400–600 nm	Rh6G/TD4 [70]; photosens [71,72,73]; porphyrazine [74]	0.8–15.7%	Phase-induced release; sustained release effect; pH dependent
		0.8–5 μm	Alkaline phosphatase [75]; guanine kinase [76]; doxorubicin (DOX) [77,78]; catalase [79]		
		20 and 80 nm	μRNA [80]; cisplatin [81]		
		17.9 μm	Ibuprofen; losartan potassium; metronidazole benzoate; nifedipine [82]	25–50%	Surface effects and diffusion
	Co-precipitation	113 nm	Catalase [79]; gentamicin [83]	20%	pH dependent; minimum inhibitory
Microcapsules (CaCO_3_ template)	Physical adsorption	2–3 μm	DQ−ovalbumine [84]		Good encapsulation effect and activity protection
			Lactoferrin [85]		
Heparin/CaCO_3_/CaP		112–384 μm	DOX [86]	1.4–1.9%	Concentration gradient; diffusion driving force; pH dependent; sustainable release
Cellulose-based CaCO_3_		2–3.5 μm	Lovastatin (LOV) [87]	12.5%	
Protamine sulfate and sodium poly(styrene sulfonate)/CaCO_3_		5 μm	Ibuprofen [88]	4.5%	
Cyclodextrin/CaCO_3_		4–6 µm	5-Fluorouracil; Na-L-thyroxine [89]		
ACC/poly(acrylic acid)	Co-precipitation	62 nm	DOX [90]	>9%	
Mucin/CaCO_3_		5.8 μm	DOX; aprotinin; insulin [91]	13%, 10%, 80%	Mucin content dependent; phase-induced release

**Table 2 pharmaceutics-14-00909-t002:** Alg-based hydrogels prepared by ionic and covalent crosslinking, their properties, and applications.

	Materials	Active Ingredient	Properties	Applications	Ref
Ionic crosslinking	Alg + CaCl_2_	Extracellular vesicles/ Dexamethasone/Whey protein/Bovine serum albumin	High encapsulation rate, sustained releasee, high plasticity	Enhances cardiac function and drug bioavailability; oral tissue reparation	[140,141,142,143]
	Alg + CS + SrCl_2_	Chondroitin sulfate	Lower water retention capacity	Facilitate proliferation of osteoblasts; modulate osteogenic factors	[144]
	Alg + Ag	Silver nanoparticles, BSA, tannic acid	Remote release, sensorics function, antibacterial properties	Oral delivery	[145,146]
	Alg + Pectin + CaCl_2_	Simvastatin	Sustained release	Promote angiogenesis, collagen synthesis and wound healing	[147]
	Alg + CaCO_3_ + D-Gluconic acid δ-lactone	Hyaluronic acid	Biocompatibility	Wound healing	[141]
Covalent crosslinking	Alg-Poloxamer + Silk fibroin		Porous, thermosensitive, strong mechanical feature	Cartilage tissue engineering.	[148]
	Alg + Chitosan + Gelatin	Tetracycline hydrochloride	Biodegradable	Antibacterial; wound healing	[149]
	Alg + Chitosan	Deferoxamine; BSA	Sustained release biodegradable, high mechanical properties	Delivery system; soft tissue engineering	[150,151,152]
	Alg + Sericin	Naproxen	Higher loading capacity and stability, sustainable release	Drug delivery	[153]
	Alg-norbornene Alg-furfuryl amine	Doxorubicin	Tunable porosity	Controlled drug release	[148]
	Alg	-	High toughness and electric conductivity	Antiseptic	[154]

**Table 3 pharmaceutics-14-00909-t003:** The properties, and functions of Alg-based hydrogels prepared by free radical polymerization.

Materials	Initiators	Active Ingredient	Properties	Functions	Ref
Alg-2-acrylamido-2-methyl propane sulphonic acid	Ammonium peroxodisulfate	Diclofenac sodium	pH-independent swelling	Sustained delivery	[165]
Alg + PVA	2-Acylamido-2-methylpropane-sulfonic acid	Tramadol HCl	pH-independent swelling	Controlled drug release	[166]
Alg-acrylamide	Ammonium persulfate Sodium metabisulfite	Acetaminophen drugs	pH sensitivity	Drug release	[167]
Alg-2-acrylamidoglycolic acid	Potassium peroxydiphosphate	-	Biodegradability	-	[168]
Alg-methacrylic anhydride	UV light	Hyaluronic acid	Injectable	Support stem cell chondrogenesis	[169]
Alg-glycidyl methacrylate	Ammonium persulfate	-	Porous structure Biocompatibility	Encapsulate umbilical vein endothelial cells	[170]
Alg-2-hydroxyethyl acrylate	Potassium persulfate	Bovine albumin serum 5-amino salicylic acid	pH sensitivity	High osteoblastic cell viability and proliferation	[171]

**Table 4 pharmaceutics-14-00909-t004:** Various CaCO_3_-based carriers used in drug delivery.

Delivery Types	Carriers	Drug	Properties
Oral	CaCO_3_ particles	Arginine [199]; peptide nanofiber [198]; 5-fluorouracil [200]	Satisfying drug efficacy; induced mucosal antibody responses.
	CaCO_3_-based stabilized Pickering emulsion	Vitamin D3 [201];	Satisfying stability and bioavailability
Colon	CaCO_3_ particles	5-fluorouracil (5-FU) and natural compound indole-3-carbinol (I3C) [202]	High loading capacity; sustainable release; low side effects
	CaCO_3_-Nanocellulose Films	5-Fluorouracil [203]	
	Pectin–chitin/CaCO_3_	Fosamax [204]	Adequate swelling, degradation, protein adsorption properties
Transdermal	CaCO_3_ particles	Cyanine 7 (Cy7) dye [205]; griseofulvin [206]	High loading capacity; sustainable release; low side effects; effective penetration effect
Intravenous	CaCO_3_ particles	DOX and alkylated NLG919 [207]; radionuclide ^68^Ga [208]; pluronic [209]	Enhanced chemo-immunotherapy of cancer; tomography; enhance ultrasound imaging
	Cu_2_O@CaCO_3_/HA	Cu_2_O [210]	Inhibit colorectal cancer distant metastasis and recurrence by immunotherapy
	NaGdF4/CaCO_3_	NaGdF4 [211]	Magnetic resonance imaging
Intranasal	CaCO_3_/HA/Diethylaminoethyl-dextran	Zolpidem [212]	Higher loading efficacy; increased anxiolytic effect.
	CaCO_3_/HA	Loperamide [213,214]	Reduce the pain sensitivity, high efficiency

**Table 5 pharmaceutics-14-00909-t005:** Various Alg-based hydrogel carriers used in drug delivery.

Delivery Types	Carriers	Drug	Properties	Ref.
Oral	Kefiran-Alg gel microspheres	Ciprofloxacin	pH-responsive, sustained release and antimicrobial	[218]
	Ag-Alg	Tanic acid	Small size, slow drug release, antibacterial function of Ag ions	[146]
	Ag-Alg	BSA	Remote release	[145]
	Alg hydrogel/chitosan micelles	Emodin	Small size, pH-sensitive, and sustained release	[219]
	Alg based-hydrogels	Diclofenac sodium	Semi-interpenetrating polymer network, pH-sensitive, colon-targeted, and antimicrobial	[220]
	Alg/carboxymethyl chitosan composite gel beads	Bovine serum albumin	pH-sensitive, long swelling time and sustained release	[221]
	Chitosan-Alg based microgel	polyphenols	pH-responsive, physicochemical stabilities and sustained release	[222]
Colon specific	Guar gum succinate-Alg beads	Ibuprofen	pH-dependent swelling behavior, almost no cytotoxicity	[229]
	Portulaca oleracea polysaccharide-Alg-borax hydrogel beads	5-fluorouracil	pH responsive, for colorectal cancer	[230]
	Alg-chitosan microspheres	Icariin	pH responsive and high retention in colon, anti-inflammatory, for colonic mucosal injury	[231]
	Chitosan, nanocellulose and Alg based polymeric system	5-Fluorouracil Levamisole hydrochloride	pH dependent release and enzymatic degradation, for colon cancer	[232]
Transdermal	Alg hydrogels	Vitamin D_3_	Highly porous (89.2 ± 12.5%) and biodegradable. Hemo- and cyto-compatible, for wound dressing	[233]
	Ca-Alg-PEGMA Hydrogels	Anti-TGF-β antibodies	High mechanical strength, biocompatible, and biodegradable, for wound healing	[234]
	Chitosan/Alg hydrogels	Alpha-tocopherol	Biodegradable, for skin injuries	[235]
	Gallic acid modified Alg self-adhesive hydrogel	Caffeine	Highly porous and excellent elasticity	[236]
	Alg/gum acacia hydrogels	ZnO_2_ nanoparticles	Biocompatible, antibacterial, anti-inflammatory, for wound healing	[237]
	*K*-carrageenan/Alg	Silver nanoparticles	Sustained release and antibacterial, for wound dressing	[238]
Intravenous	Fe_3_O_4_/polyethyleneimine/Alg	Fe_3_O_4_	Enhanced magnetic resonance (MR) imaging	[239]
	Glycyrrhizin/Alg	Antioxidant quercetin	Improved liver targeting and therapeutic efficacy	[240]
Intranasal	Alg	Ropinirole hydrochloride	High encapsulation efficiency, safe to nasal epithelium.	[241]
		Lipopolysaccharide	Induce an effective systemic and mucosal immune response	[242]
	Alg/CS	PPE17 antigen	Induce strong immune response	[243]

**Table 6 pharmaceutics-14-00909-t006:** Summary of comparison of CaCO_3_, Alg, CaCO_3_/Alg carriers.

	CaCO_3_ (ACC and Vaterite)	Alg		CaCO_3_/Alg	
		Alg-Hs	Hollow Capsules	Alg Coated on CaCO_3_	CaCO_3_ Embedded in Alg
Method	Facile, low-cost, eco-friendly, efficient, accessible	Facile, diverse, malleable, accessible		Facile, low-cost, eco-friendly, efficient, diverse, accessible	
Size	About 30 nm [90]–20 um [42]	Nano to millimeter [271]	Determined by matrix	Determined by CaCO_3_	Determined by Alg
Shape	Spherical and elliptical [33]	Inconstant			
Mechanism properties	Relatively high strength	Determined by the ratio of M/G [29], softer than CaCO_3_		Harder than pure Alg	
Loading capacity	Large molecules: up to 25% [83] Small molecules: up to 80% [91]	Relatively lower than CaCO_3_ [195]		Higher than CaCO_3_ and Alg, high encapsulation efficiency (up to 90%) [272]	
Release	pH dependent [93]	Interface and concentration dependent [195]		Synergistic effect; sustaining release [115]	
Application	Diverse drug delivery system; tissue engineering [273]	Diverse drug delivery system, wound healing, tissue engineering [29]		Diverse drug delivery system, tissue engineering, wound healing, bone regeneration [274,275,276,277,278,279,280]	

**Table 7 pharmaceutics-14-00909-t007:** Hybrid carriers based on a combination of CaCO_3_ and Alg, their properties and applications.

Applications	Carriers	Drugs or Test Objects	Properties
Drug delivery	CaCO_3_/Alg	Sorafenib tosylate/ Gold nanoparticles [275], Lysozyme [277], Peptide [285], Phages [281]	High drug loading efficiency, controlled and sustainable release, high cell inhibition rate;
	CaCO_3_/Alg+ PUA/PAA	Indomethacin [278]	
	Alg/Polyquaternium-10 (CaCO_3_ template)	Rhodamine B and 6-hydroxy fluorescein molecule [280]	
	Alg/PSS/PAH (CaCO_3_ template)	Rhodamine-labeled BSA [286]	
Tissue engineering	CaCO_3_/Alg	Alkaline phosphatase [274]	Support osteoblast growth and new tissue formation, good cell adhesion and proliferation ability
	Alg scaffolds (CaCO_3_ template)	Dextran [279]	
Vascular imaging, ultrasound (US) imaging	CaCO_3_/Alg	BaSO_4_ [276]; Pluronic [209]	Good medical perfusion and angiography effects

## Data Availability

Not applicable.

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
