# Peer review of "Hard, Soft, and Hard-and-Soft Drug Delivery Carriers Based on CaCO3 and Alginate Biomaterials: Synthesis, Properties, Pharmaceutical Applications"

_pharmaceutics, 2022, doi:10.3390/pharmaceutics14050909_

Round 1

Reviewer 1 Report

The present manuscript presents an overview of the literature regarding the use of CaCO3 particles and alginate hydrogels for drug delivery purposes. The manuscript could be published, nevertheless I have some comments.

1) It is not clear to the reader why the authors have chosen those specific materials for the present review. Especially since the first one is an inorganic mineral and the second one an organic polymer, it is not very probable readers will be interested in both CaCO3 and alginate drug delivery. Finally, few works combining those two materials are reported and commented.

2) I suppose the present review is not an exhaustive review. The tables that summarize some reported works are interesting. Nevertheless, it is not clear how the authors have chosen the specific works that are presented (novelty? originality? good results? recent publications?).

3) Some publications are presented in sections 2.3 and 3.3 where drug release results are discussed. It is not clear how these publications differ from the ones presented in section 4.

4) Lines 103-104 vs Figure 2. It is not clear which processes from Figure 2 correspond to route a) or b).

5) The meaning of lines 114-117, 171-173 and 177-180 is not very clear, these sentences need rephrasing or some additional explanation.

6) The term “non-symmetrical reaction of ions” is confusing.

7) Lines 191-192: “showed that a rapid release of 84% and 45% occurred at pH =7.4 (blood physiological) and 4.8 in 191 the first 3 h”. I guess the order of the two percentages has probably been inversed.

8) Lines 219-221, the authors should add a justification regarding the low release at pH 5.5.

9) Line 337, how are the antibacterial properties connected to the crosslinking method?

10) Line 254 it is reported that CaCO3 are good sacrificial templates but almost no such examples are commented within the text.

11) In section 2.3, it would be useful if the authors could present comparatively the advantages and drawbacks of each drug loading method.

12) In Figure 6, figures A and D are not readable (notably the chemical structures). The resolution of Figures 7A, 7C and 7D is not satisfactory.  

13) There are some typos and other mistakes in the text. For instance, polyethylene glycol in line 57, line 68 poly (lactic acid) (twice), table 1 poly (acrylic acid). In general, the names of the polymers should be checked, there shouldn’t be any gap between “poly” and the “(“. Lines 109 and 113 to synthesis, line 115 those specific gas, line 136 meat, line 273 for four times, line 456 was existed, etc.

Author Response

Q1) It is not clear to the reader why the authors have chosen those specific materials for the present review. Especially since the first one is an inorganic mineral and the second one an organic polymer, it is not very probable readers will be interested in both CaCO3 and alginate drug delivery. Finally, few works combining those two materials are reported and commented.

Response: Supplements have been made in the paragraph of “introduction” to make it clearer why we choose these two materials for review, also, Figure 1 was added to make the text easier to understand. (On page 1-4). we describe CCPs and Alg-Hs as representatives of two opposite classes of materials (inorganic and organic or hard versus soft, respectively) highlighting their advantages and disadvantages, similarities, and differences. We look at them as both stand-alone carriers and a combination of a hybrid carrier combining these materials. The analysis here is made looking at the intrinsic properties to the preparation methods (green and facile) and performance in biological and pharmaceutical applications, and they have been widely studied as focal points. In addition, the cross-linking of Alg can be induced by Ca2+, and the diverse combination of CCPs and Alg can form hybrid carriers with coatings, hollow Alg capsules as well as CCPs-enhanced Alg hydrogels, all of which make these two materials more attractive. Therefore, this review summarized the properties, synthetic methods, and recent studies of CCPs and Alg-Hs as carriers for drug delivery and targeted therapy. Moreover, the drug loading efficiency, release mechanism and applications through different administration modes of the two carriers are emphasized and compared, various combinations of the two materials and their application scenarios are described, which provide insights for the improvement of the performance of drug delivery system and the exploitation of new carriers.

In addition, we have further supplemented the studies on the combination of the two materials in section 5 of the text, and listed relevant examples in Table 7 and added Figure 10. (On page 22-24).

Q2) I suppose the present review is not an exhaustive review. The tables that summarize some reported works are interesting. Nevertheless, it is not clear how the authors have chosen the specific works that are presented (novelty? originality? good results? recent publications?).

Response: More consistent examples were added with a focus on loading, release part. (On page 6-10, 14-15) and application part (On page 20-21). Also, we have supplemented the content about the comparison and combination of the two materials in this text (On page 22-23) and Table 6 and 7. We mainly selected the specific works on CaCO3 and alginate hydrogels in the field of drug carriers in the past decade, which can be noticed from the publication dates of the references.

Q3) Some publications are presented in sections 2.3 and 3.3 where drug release results are discussed. It is not clear how these publications differ from the ones presented in section 4.

Response: We reorganized materials in sections and splits the loading release and applications in different sections. First, the application of the carriers was classified by delivery mode in section 4, the main content was about the toxicology and drug efficacy of the drug carrier in vitro and in vivo, which obviously related to some classic drugs. The prerequisite for the application of drug carriers is the testing of drug loading and release properties, which inevitably involves the selection of some easy-to-detect and low-cost drugs or dyes for experiments, and some publications only explore the basic properties of the carrier without conducting follow-up in vivo experiments. Therefore, some typical publications containing systematic research can be discussed in both two parts with different emphasis. But more is to select different publications to summarize. Moreover, in order to make the logic of the article clearer and more readable, and prevent the content from being cluttered, it is a good choice for some publications containing more comprehensive research to be discussed separately in two different sections.

Q4) Lines 103-104 vs Figure 2. It is not clear which processes from Figure 2 correspond to route a) or b).

Response: Routes a and b represent different synthesis conditions, but Figure 2 is a summary of the formation mechanism involved in the synthesis of CaCO3 through different routes. The supplementary content was added to illustrate each of the routes in Figure 2in the text. (On page 5)

Q5) The meaning of lines 114-117, 171-173 and 177-180 is not very clear, these sentences need rephrasing or some additional explanation.

Response: We have rephrased these sentences in the text. However, it should be noted that the original sentence may no longer exist because we have made lots of revisions and summaries of this part in the text, please review the corresponding part of the text. (Line 170-173, Line 226-234, Line 316-320)

Q6) The term “non-symmetrical reaction of ions” is confusing.

Response: “non-symmetrical reaction of ions” refers to the selection of different concentrations of Ca2+ and CO32- solutions for mixing, even if Ca2+ ions combine with CO32- ions according to the ratio of 1:1 to form CaCO3 particles. However, the morphology of CaCO3 particles will be ultimately altered to anisotropy due to the presence of an excess of ions (Ca2+ or CO32-) around the nuclei.

Q7) Lines 191-192: “showed that a rapid release of 84% and 45% occurred at pH =7.4 (blood physiological) and 4.8 in 191 the first 3 h”. I guess the order of the two percentages has probably been inversed.

Response: After re-checking the corresponding literature, we confirm that the order of the two percentages has been inversed and we have corrected them in the text. (Line 322-323)

Q8) Lines 219-221, the authors should add a justification regarding the low release at pH 5.5.

Response: The reasons for the low release rate at pH 5.5 have been explained on page 10 in the text. (Line 350-357)

Q9) Line 337, how are the antibacterial properties connected to the crosslinking method?

Response: Metal particles such as silver, zinc, and strontium, etc. are good choices for improving the antibacterial properties of composite materials, so they are often selected and doped into hydrogel materials. Given the volatile electronic properties of metals and the multifunctional groups of polymers, ionic crosslinking is a feasible method to efficiently dope metal particles into capsules. Appropriate data were added to table 5.

Q10) Line 254 it is reported that CaCO3 are good sacrificial templates but almost no such examples are commented within the text.

Response: Thank you for your valuable comments. We have supplemented in section 2.3.3 (On page 9-10), section 5 (On page 22-23) and Table 7 in the text.

Q11) In section 2.3, it would be useful if the authors could present comparatively the advantages and drawbacks of each drug loading method.

Response: we have made supplements on page 7 to compare these two methods in the text. It should be noted that drug molecules loaded through physical adsorption are maintained by electrostatic force, and the adsorption efficiency is highly dependent on the characteristics of surface of both drug molecules and CCPs. Those molecules adsorbed by weak electrostatic forces will inevitably be desorbed and lost during the washing process, which will decrease the loading efficiency of carriers. But this method can effectively maintain the original properties of drug molecules or the activity of protein, and less resistance is required for the release of molecules [84]. In co-precipitation method, the loading of molecules is driven by similar electrostatic interactions, and high loading efficiency can be obtained since molecules will be trapped in the voids during the formation process of CCPs, and there is lower leakage efficiency of carriers, even though a small number of drug molecules with small size will be missed [84]. However, the alkaline environment during particle formation process may reduce the activity of drugs, especially proteins, thus reducing drug efficacy [85]. Therefore, the polyelectrolyte film formed by layer-by-layer assembly (LbL) technique combined with the above loading method will provide an effective means for drug encapsulation.

Q12) In Figure 6, figures A and D are not readable (notably the chemical structures). The resolution of Figures 7A, 7C and 7D is not satisfactory.

Response: Thank you for your advices, we have improved the quality and resolution of all figures, and replaced the original figures in the text.

Q13) There are some typos and other mistakes in the text. For instance, polyethylene glycol in line 57, line 68 poly (lactic acid) (twice), table 1 poly (acrylic acid). In general, the names of the polymers should be checked, there shouldn’t be any gap between “poly” and the “(“. Lines 109 and 113 to synthesis, line 115 those specific gas, line 136 meat, line 273 for four times, line 456 was existed, etc.

Response: We are very sorry for the negligence of such details, we have carefully read and revised the whole article, and the revised parts have been tracked.

In addition, we have made changes to other sections of the text. For example, splitting the original "loading and release" to make it more readable. 4.5 Intravenous drug delivery and 4.6 Intranasal drug delivery were added in the application part, and in section 5, we added the comparison and combination of the two materials; Anyway, the revised parts of the text have been marked in red;

! We also added Figures 1, 4, 5 and 10 on the basis of the original draft to make the structure of the text clearer and more complete.

Reviewer 2 Report

Line 68: Paragraph begins with the phrase “those polymeric particles”. What are the polymeric particles to which the authors refer to?

Line 136: probably the authors wanted to write “meant” instead of “meat”.

Line 151: The way “Those carriers with the core-shell structure” is written, seems like all CCPs have core-shell structure, which is not correct.

Author Response

Q1) Line 68: Paragraph begins with the phrase “those polymeric particles”. What are the polymeric particles to which the authors refer to?

Response: We are sorry that we didn't express the meaning of this sentence clearly. We have corrected that part, strictly speaking, the 'introduction' part.

At present, a large number of drug delivery carriers with favorable biological properties have been developed, including among others two main classes: inorganic and organic carriers. The former one, inorganic particles, include among others such carriers as quantum dots (QDots), gold (Au) and silver (Ag) metal nanoparticles, clays, carbon, porous silica (SiO2), hydroxyapatite, and calcium carbonate (CaCO3). The latter one, organic particles, comprise of not less impressive range of carriers such as liposomes, polymer micelles, dendrimers, cell penetrating peptides, protein conjugates and alginate (Alg) (Figure 1). (Line 30-35)

Q2) Line 136: Probably the authors wanted to write “meant” instead of “meat”.

Response: We checked the full text and corrected any possible spelling errors, and each correction was marked.

Q3) Line 151: The way “Those carriers with the core-shell structure” is written, seems like all CCPs have core-shell structure, which is not correct.

­Response: We have removed the word "those" to avoid ambiguity. In addition, "The carriers with core-shell structure" refers to a class of carriers with this characteristic, which is beneficial to the loading and release of drugs, rather than CCPs in particular. In addition, as you mentioned, not all CCPs have core-shell structure, which has also been mentioned in composite carriers with surface coating formed by ACC and vaterite as matrix in the later part of this paragraph (Line 287-290).

! In addition, we have made changes to other sections of the text. For example, splitting the original "loading and release" to make it more readable. 4.5 Intravenous drug delivery and 4.6 Intranasal drug delivery were added in the application part, and in section 5, we added the comparison and combination of the two materials; Anyway, the revised parts of the text have been marked in red;

! We also added Figures 1, 4, 5 and 10 on the basis of the original draft to make the structure of the text clearer and more complete.

Reviewer 3 Report

The manuscript entitled “Inorganic CaCO3 and organic alginate -based drug delivery carriers: comparison of synthesis, properties, and applications” reported the method of synthesis and the properties of Calcium carbonate and Alginate-based hydrogel carriers as well as their applications. The manuscript is well written and the authors have exerted a good effort in addressing the data. Few issues must be considered before publication.

  • Lines 28-29; please rephrase the sentence as “suitable cells, pathogens and versatile particles” are not abundant drug delivery carriers as liposomes, polymer micelles”.
  • Figures should stand alone. Abbreviations should be mentioned as a footnote to each figure.
  • Some sentences need rephrasing as it is not scientifically correct to start a sentence with the word “And”. Kindly refers to lines 153, 343
  • In table 2; authors mentioned “alginate and chitosan” in both ionic crosslinking and covalent crosslinking. Please discuss this in more details.
  • Some typo mistakes need correction. Line 519 “mL-1”, line 560 “though”, line 634 “resent”.
  • Insert citation before (Figure 7C).

Author Response

Q1) Lines 28-29; please rephrase the sentence as “suitable cells, pathogens and versatile particles” are not abundant drug delivery carriers as liposomes, polymer micelles”.

Response: We have exchanged this sentence to “At present, a large number of drug delivery carriers with favorable biological properties have been developed, including among others two main classes: inorganic and organic carriers. The former one, inorganic particles, include among others such carriers as quantum dots (QDots), gold (Au) and silver (Ag) metal nanoparticles, clays, carbon, porous silica (SiO2), hydroxyapatite (HAP), Ca3(PO4)2 and calcium carbonate (CaCO3). The latter one, organic particles, comprise of not less impressive range of carriers such as liposomes, polymer micelles, dendrimers, protein conjugates, red blood cells (RBCs), cell penetrating peptides and alginate (Alg) (Figure 1).” (Line 30-35)

Q2) Figures should stand alone. Abbreviations should be mentioned as a footnote to each figure. Some sentences need rephrasing as it is not scientifically correct to start a sentence with the word “And”. Kindly refers to lines 153, 343

Response: We inserted figures into the text as requested by the journal rather than at the end of the article, and we also uploaded a zip file of figures separately when we submitted the manuscript, which I thought would be helpful. In addition, all abbreviations mentioned in the figures have been explained in parentheses, and we have corrected any sentences that needed to be improved in the text.

Q3) In table 2; authors mentioned “alginate and chitosan” in both ionic crosslinking and covalent crosslinking. Please discuss this in more details.

Response: Alg solution is converted to hydrogel by ionotropic gelation with polyvalent metal ions, indicating that the gelation process involves the participation of ions. Covalent cross-linking relies on copolymerization or polycondensation of the functional groups of polymers themselves, rather than ion induction, which has been described in detail in this paper. Therefore, after further reading of relevant literature, although the ions interaction was mentioned in highlight part in relative reference, the hybrid hydrogels formed by alginate and chitosan tend to be obtained through covalent cross-linking between organic compounds, so we have made modifications in the table.

Q4) Some typo mistakes need correction. Line 519 “mL-1”, line 560 “though”, line 634 “resent”. Insert citation before (Figure 7C).

Response: We have corrected the corresponding positions and inserted citations before Figure 7C in the text.

! In addition, we have made changes to other sections of the text. For example, splitting the original "loading and release" to make it more readable. 4.5 Intravenous drug delivery and 4.6 Intranasal drug delivery were added in the application part, and in section 5, we added the comparison and combination of the two materials; Anyway, the revised parts of the text have been marked in red;

! We also added Figures 1, 4, 5 and 10 on the basis of the original draft to make the structure of the text clearer and more complete.
